# Strontium isoscape of sub-Saharan Africa allows tracing origins of victims of the transatlantic slave trade

Strontium isotope ($^{87}$Sr/$^{86}$Sr) analysis with reference to strontium isotope landscapes (Sr isoscapes) allows reconstructing mobility and migration in archaeology, ecology, and forensics. However, despite the vast potential of research involving $^{87}$Sr/$^{86}$Sr analysis particularly in Africa, Sr isoscapes remain unavailable for the largest parts of the continent. Here, we measure the $^{87}$Sr/$^{86}$Sr ratios in 778 environmental samples from 24 African countries and combine this data with published data to model a bioavailable Sr isoscape for sub-Saharan Africa using random forest regression. We demonstrate the efficacy of this Sr isoscape, in combination with other lines of evidence, to trace the African roots of individuals from historic slavery contexts, particularly those with highly radiogenic $^{87}$Sr/$^{86}$Sr ratios uncommon in the African Diaspora. Our study provides an extensive African $^{87}$Sr/$^{86}$Sr dataset which includes scientifically marginalized regions of Africa, with significant implications for the archaeology of the transatlantic slave trade, wildlife ecology, conservation, and forensics.

In Africa, human mobility and migration played fundamental roles in the evolution of our species[1] and formed the diversity of peoples, cultures, and linguistics of the continent[2,3]. For instance, the great expansion of Bantu-speaking agriculturists from West Africa to eastern and southern Africa starting ~4000 years ago significantly contributed to the dynamics of contemporary African populations, subsistence practices, and languages[4]. One of the most notable migration events in the history of the continent occurred with the transatlantic slave trade between the 15$^{th}$ and 19$^{th}$ centuries, during which at least 12.5 million Africans were abducted, enslaved, and transported to the Americas and Europe, which dramatically changed the demographics, economics, and politics of both Africa and the Americas[5]. Although the transatlantic slave trade is well documented and known as the largest forced migration event in human history[5], archaeologists and historians alike have struggled to identify the geographic origins and individual life histories of enslaved individuals in the African Diaspora. Countless historic documents describe the journeys of at least 36,079 vessels transporting African captives and include detailed information on the African ports visited, and the total number of captive individuals embarked on a given ship[5]. However, the individual treacherous

journeys that captive Africans endured before reaching coastal shipping ports, as well as the regions from which they were taken, remain largely unclear in the archaeology of the African Diaspora. Recently, ancient DNA (aDNA) and isotopic analyses have been successfully applied to archaeological human remains from sites associated with the slave trade in the Caribbean[6–8], Brazil[9], North America (e.g., refs. 10–12), St Helena[13], and South Africa[14], providing critical first insights into the African origins of enslaved individuals. While information from aDNA can potentially determine the population ancestry of an individual, it will not illuminate where a person was born and raised[15]. A growing number of such studies have employed strontium isotope ($^{87}$Sr/$^{86}$Sr) analysis of human remains, which demonstrates great potential for reconstructing the geographic origins and migration of individuals, particularly when local- or large-scale strontium isoscapes are available as references[16–18].

The $^{87}$Sr/$^{86}$Sr ratios of a location primarily relate to the underlying bedrock geology, with additional influences from atmospheric deposition geomorphological, and biochemical processes, and remain stable over archaeological time scales[19]. $^{87}$Sr/$^{86}$Sr ratios in animal and human body tissues primarily mirror biologically

✉ e-mail: voelze@ucsc.edu

available $^{87}Sr/^{86}Sr$ incorporated via the consumption of plants and water from the local substrate with minimal fractionation which is corrected during post-measurement data processing[16,17] (Supplementary Note 1). In African archaeological contexts, $^{87}Sr/^{86}Sr$ analysis has allowed the tracing of the Paleolithic ostrich eggshell bead trade[20] and the landscape use of early hominins[21]. The analysis of $^{87}Sr/^{86}Sr$ can also be applied in the study of wildlife ecology and conservation. Sub-Saharan Africa harbours some of the largest global biomass movements related to seasonal migrations of mammals, birds, and insects[22]. Previous $^{87}Sr/^{86}Sr$-based studies have provided critical insights into the habitat use of African elephants[23] and rhinoceroses[24], as well as forensic studies of the ivory trade[25]. Other forensic applications could include the identification of human origins in forensic cases, such as the remains of deceased migrants[26]. However, the vast potential of $^{87}Sr/^{86}Sr$ analysis in African archaeology, ecology, and forensics remains largely unexplored as Sr isoscapes do not yet exist for the largest parts of the continent. In particular, the absence of data from West and West-Central Africa impedes our ability to use $^{87}Sr/^{86}Sr$ analysis in the study of slavery and the identification of the specific geographic origins of the countless victims of the slave trade.

In the last two decades, strontium isoscapes have been successfully developed at the local, regional, and even global scale using environmental and/or archaeological samples and various modelling approaches (Supplementary Note 1)[17,27]. The current leading approach in isoscape modelling employs random forest (RF) regression, which integrates georeferenced $^{87}Sr/^{86}Sr$ data with a range of geological and environmental covariates affecting natural isotopic variation and displays high accuracy in their spatial prediction of $^{87}Sr/^{86}Sr$[17,18]. Numerous strontium isoscapes using RF modelling methods are now available for Europe[18,28], North America[29], New Zealand[30], Madagascar[17], parts of Tanzania and Kenya[31], Angola[32], and even globally[17]. However, regions with very limited sample coverage continue to exhibit poor spatial $^{87}Sr/^{86}Sr$ predictions in the global Sr isoscape, particularly in Africa. This research gap in Africa can be attributed to the presumably high cost of large-scale $^{87}Sr/^{86}Sr$ projects and the lack of the respective isotope ratio mass spectrometer infrastructure in sub-Saharan Africa, with the exception of South Africa. In addition, obtaining samples in many African regions is challenging due to considerable logistical difficulties, as well as conflicts and instability (e.g., Burkina Faso, Burundi, the Central African Republic, Mali, Sudan, South Sudan, Somalia, and parts of Nigeria[33]).

This study aims to draft a strontium isoscape of sub-Saharan Africa using both newly measured and previously reported bioavailable $^{87}Sr/^{86}Sr$ data. We analyze 778 environmental samples (including plants, soil leachates, and microfauna) from 24 African countries, focusing particularly on West and West-Central Africa, where $^{87}Sr/^{86}Sr$ data were previously nearly absent (Fig. 1b). We combine this dataset with the 1488 published bioavailable $^{87}Sr/^{86}Sr$ data (Supplementary Data 1) and employ a random forest modelling approach to generate a bioavailable strontium isoscape for sub-Saharan Africa. By applying this isoscape to $^{87}Sr/^{86}Sr$ data from enslaved Africans at two cemeteries in the African Diaspora, namely the Anson Street African Burial Ground in the US[10] and the Pretos Novos cemetery in Brazil[9], we demonstrate the predictive potential of this isoscape in the bioarchaeology of slavery. To aid in the estimation of individual origin, we combine published human tooth enamel $^{87}Sr/^{86}Sr$ data with corresponding genetic evidence for the Anson Street African Burial Ground[34] or oxygen isotope data for the Pretos Novos cemetery[35]. From both cemeteries, we select five individuals whose $^{87}Sr/^{86}Sr$ ratios are inconsistent with their regions of captivity and burial. Finally, we discuss the potential further applications of this Sr isoscape in wildlife ecology and forensics in sub-Saharan Africa.

## Results and discussion

### Bioavailable $^{87}Sr/^{86}Sr$ variability in sub-Saharan Africa

The bioavailable $^{87}Sr/^{86}Sr$ ratios in sub-Saharan Africa range from 0.70381 to 0.87810, with a mean of $0.71774 \pm 0.01229$ ($1\sigma$), reflecting the diverse geological characteristics of the African continent (Fig. 1a and Supplementary Note 2). In addition to the large isotopic variability, Africa exhibits many highly radiogenic $^{87}Sr/^{86}Sr$ compositions, with a first quartile to third quartile (Q1–Q3) range of 0.7099–0.7223. This range far exceeds that reported for any other continent, including Europe (Q1–Q3 of 0.7087–0.7120)[17], the USA (Q1–Q3 of 0.7088–0.7109)[17], China (Q1–Q3 of 0.7096–0.7119)[36], and even worldwide (Q1–Q3 of 0.7084–0.7115)[17]. Notably, the most radiogenic $^{87}Sr/^{86}Sr$ ratios ($> 0.730$) are restricted to regions dominated by underlying Archaean bedrock, such as in present-day Angola, Zimbabwe, Zambia, western Tanzania, northern South Africa, and several southern West African countries (Supplementary Note 3).

### Strontium isoscape modelling results

We selected environmental variables based on extensive research from previous isoscape studies[17,18] and excluded those that might lead to correlation and collinearity issues within our model. Ultimately, we used 11 predictors (dust, terrane age, soil clay content, soil cation exchange capacity, soil organic carbon content, maximum age of bedrock, sea salt deposition, mean annual precipitation, srsrq3, lithology, and elevation) to model the $^{87}Sr/^{86}Sr$ ratios of 2266 samples using a random forest algorithm tuned with both hyperparameters (number of variables randomly sampled at each split and maximum node size) set to 2. Overall, the RF model explained 80% of the out-of-bag variance, with an average root-mean-square error (RMSE) of 0.0056. Approximately 63% of the model residuals fell within the range of −0.002 to +0.002 (Supplementary Fig. 1). We detected no spatial autocorrelation in the model residuals, indicating that no unaccounted proximity effects (between samples) existed within the variance unexplained by the model (Supplementary Fig. 2). The RF model demonstrated strong predictive performance on our validation set ($R^2 = 0.84$, RMSE = 0.0056), enabling us to use the fitted model to predict the pattern of $^{87}Sr/^{86}Sr$ variation across sub-Saharan Africa (Fig. 2a). At least 75% of the standard predicted errors of the isoscape fell within the range of 0.0016–0.0075 (Fig. 2b).

Our Sr isoscape intentionally excludes regions where at least one environmental predictor variable value falls outside the range of those represented by our $^{87}Sr/^{86}Sr$ sampling (grey regions in Fig. 2c). This conservative depiction of the isoscape accounts for the fact that RF models are not reliable when extrapolating into areas without matching training data. In addition, we provide a standard error map to represent prediction accuracy (Fig. 2b) and multivariate Mahalanobis distances to indicate regions of environmental dissimilarity (Fig. 2c).

The ranking of the relative importance of input parameters indicates that dust deposition is the most important factor influencing spatial bioavailable $^{87}Sr/^{86}Sr$ variation, followed by terrane age, soil clay content, maximum age of bedrock, soil cation exchange capacity, sea salt deposition, mean annual precipitation, elevation, soil organic carbon content, srsrq3, and lithology (Supplementary Fig. 1). The partial dependence plots (Supplementary Fig. 3) reveal more radiogenic $^{87}Sr/^{86}Sr$ ratios occur in regions with low dust deposition, while lower and relatively homogeneous $^{87}Sr/^{86}Sr$ ratios are more prevalent in regions receiving a high annual amount of aeolian dust, such as central southern Africa associated with the Kalahari Desert. Geological variables also dominate in predicting bioavailable $^{87}Sr/^{86}Sr$ across Africa, including the age of geological units and soil properties, both of which have nearly a linear effect on bioavailable $^{87}Sr/^{86}Sr$ ratios. We observe more radiogenic $^{87}Sr/^{86}Sr$ ratios in several craton regions dominated by Archaean plutonic and metamorphic rocks, which tend to form soils with low cation

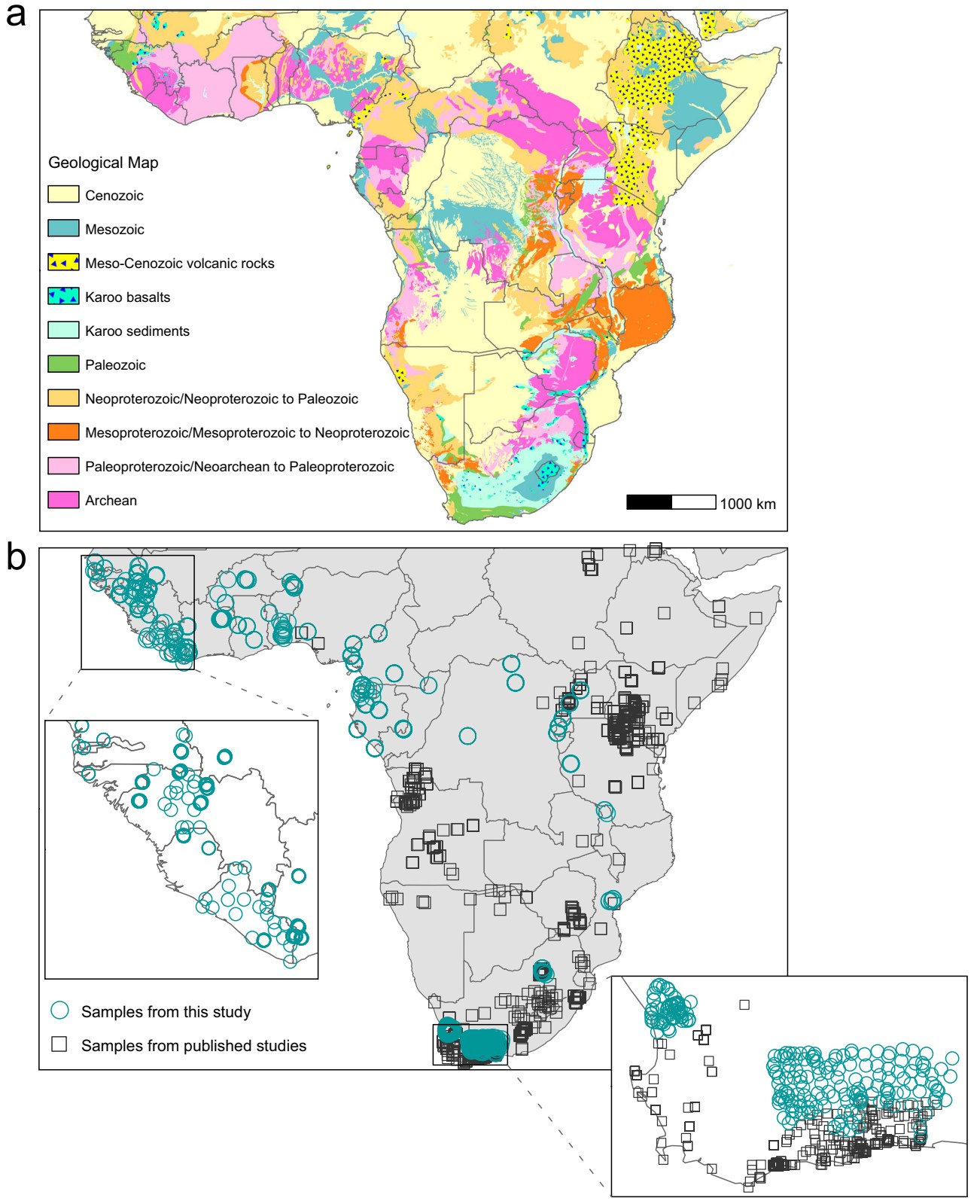

**Fig. 1 | Geological map and sampling locations. a** Simplified geological map, modified after ref. 90 using ArcGIS Desktop 10.8. **b** Map showing the environmental sampling locations from this study and previously published work. The sampling locations focused on filling gaps in West Africa, West-Central Africa, and parts of South Africa, covering all major geological units across the African continent south of the Sahara.

exchange capacity and clay content[31,37]. Conversely, low $^{87}Sr/^{86}Sr$ ratios are found in regions covered by Mesozoic-Cenozoic volcanic rocks in East Africa and basalts in central South Africa, where soils generally have high cation exchange capacity and clay content[31,37]. In addition, our $^{87}Sr/^{86}Sr$ data show a positive relationship with both mean annual precipitation and elevation, with the former possibly influenced by increased silicate weathering rates due to higher precipitation and the latter resulting from the exposure and

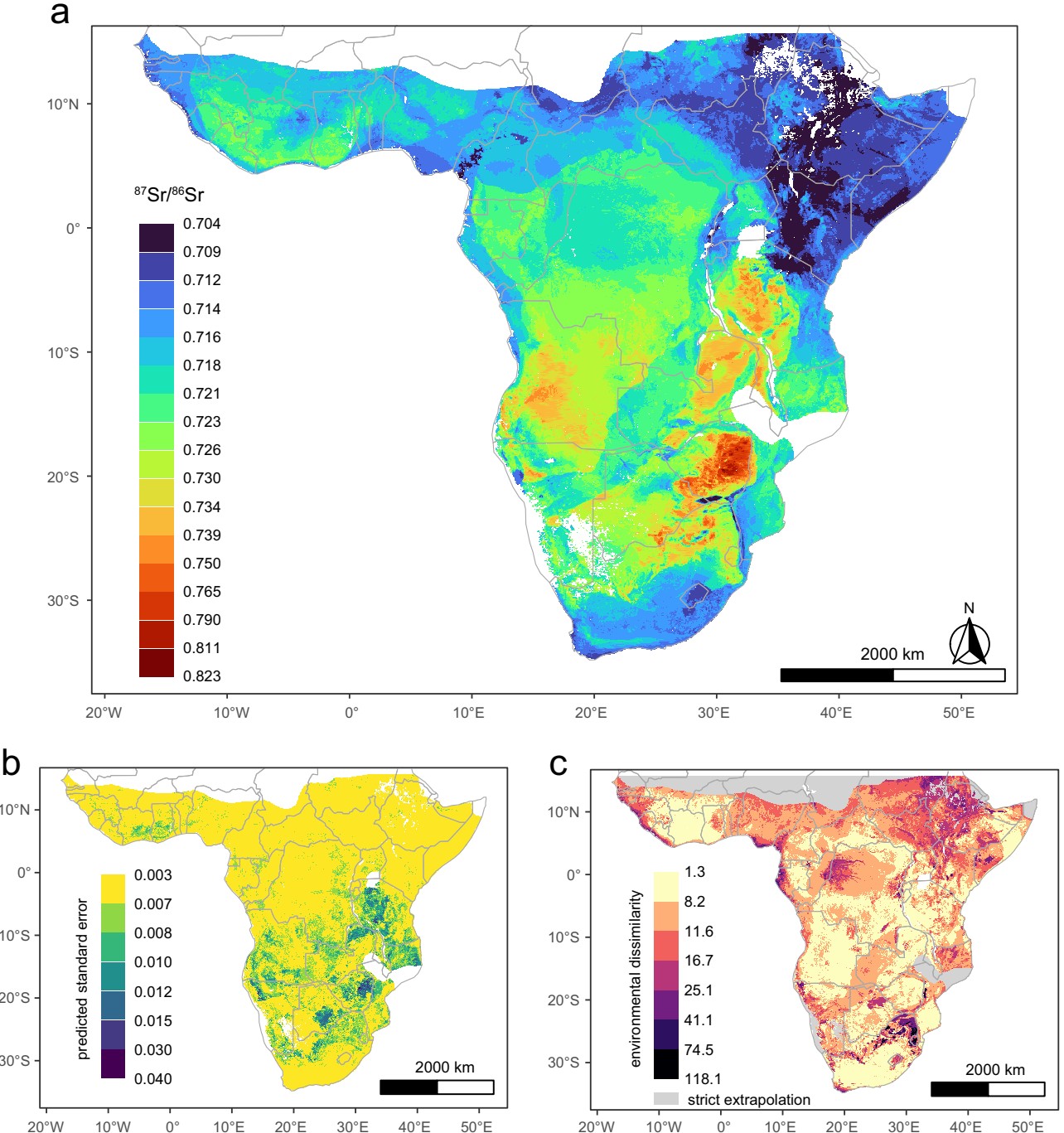

**Fig. 2 | Bioavailable strontium isoscape and associated standard error map for sub-Saharan Africa. a** Bioavailable strontium isoscape for sub-Saharan Africa. **b** Standard error map for the strontium isoscape. **c** Mobility-oriented parity metrics. The multivariate Mahalanobis distance, or environmental similarity metric, relates to the comparison between the calibration dataset and prediction regions[70]. Darker-shaded areas indicate greater dissimilarity from the environmental conditions on which the model was trained, suggesting that using the isoscape for these regions requires greater caution. All maps use non-linear natural breaks in the colour scales and their corresponding legends. White or grey-shaded areas indicate regions with strict extrapolation of environmental predictor variables, where at least one predictor variable value is not represented by $^{87}$Sr/$^{86}$Sr data from elsewhere, hindering reliable predictions of local $^{87}$Sr/$^{86}$Sr ratios in these areas.

weathering of older bedrock during mountain building processes and increased physical erosion in high relief areas[38]. Other predictors show comparatively weak associations with the observed $^{87}$Sr/$^{86}$Sr ratios.

### Geographic assignment results

The isotopic geolocation of five individuals from the Anson Street African Burial Ground aligns well with previous genetic assessments.

Two individuals of West-Central African ancestry correspond relatively well to the eastern-central parts of Angola (Fig. 3b, c). The individuals identified as having West African ancestry show distinct geographic assignment results. The $^{87}$Sr/$^{86}$Sr ratio of one individual is particularly prevalent in West Africa and the assignment matches large regions of present-day Liberia, Côte d'Ivoire, Guinea, Sierra Leone, and Mali (Fig. 3e). The more radiogenic $^{87}$Sr/$^{86}$Sr ratios of two other individuals are less common in West Africa but occur within a 100 km stretch

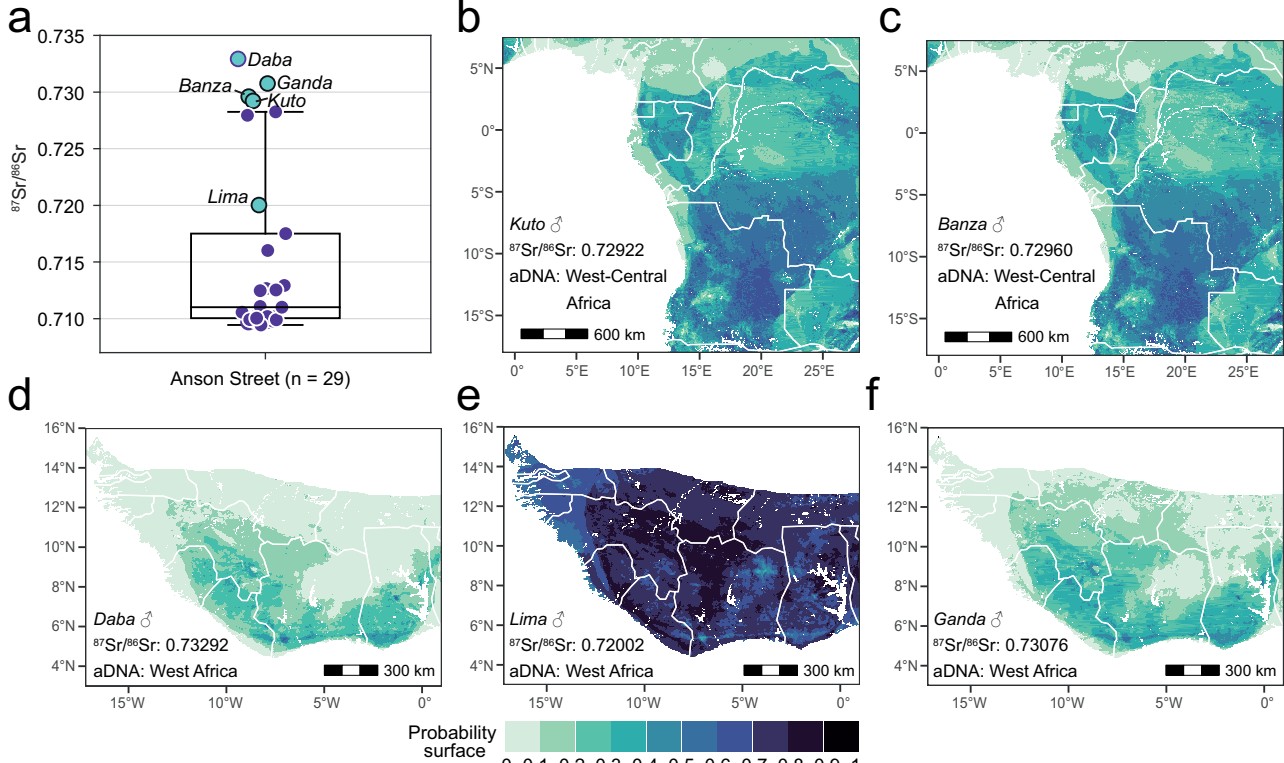

**Fig. 3 | Normalised probability of geographic origins for five enslaved individuals excavated from the Anson Street African Burial Ground in Charleston (USA), using published tooth enamel $^{87}Sr/^{86}Sr$ data and our strontium isoscape, in combination with aDNA data restricting the potential regions of origin[10].**
**a** Boxplot depicting the distribution of $^{87}Sr/^{86}Sr$ data for 29 individuals buried at the site (Supplementary Data 3), generated using OriginPro 2021 software. The box represents the interquartile range (IQR) from the 25th to the 75th percentile, with a line indicating the median. Whiskers extend to the minimum and maximum values within 1.5 times the IQR from the quartiles, with outliers defined as points beyond this range. Lower $^{87}Sr/^{86}Sr$ ratios between 0.709 and 0.715 are likely present in coastal South Carolina, where the Anson Street Ancestors were held captive and buried[10,34]. Based on this, five individuals (marked with green symbols) with well-preserved aDNA and $^{87}Sr/^{86}Sr$ ratios ≥ 0.720 were selected for isotopic geolocation modelling. **b, c** shows the isotopic geolocation to parts of West-Central Africa for the individuals named *Kuto* and *Banza*, respectively. **d–f** shows the isotopic geolocation of the individuals named *Daba*, *Lima*, and *Ganda* within West Africa. The probability surface was normalised by dividing each cell's value by the maximum probability. Consequently, darker regions represent areas with a higher relative probability of origin compared to other cells, rather than the actual probability values.

along the southern coast of West Africa and in the eastern provinces of Guinea (Fig. 3d, f).

The isotope-based geographic assignments for five individuals from the Pretos Novos cemetery, using a combination of $^{87}Sr/^{86}Sr$ and $δ^{18}O$ data, suggest that four individuals (P3, P5, P13, and P16) could have originated from different regions within Angola or parts of South-East Africa, as both isotope systems show similarity between these regions (Fig. 4c–f). The lower $^{87}Sr/^{86}Sr$ ratio of individual P6 is distinct and aligns with isoscape data from parts of West Africa (Guinea and Nigeria), Cameroon, and South Africa (Fig. 4b).

## A strontium isoscape of sub-Saharan Africa applied to African Diaspora sites

This study presents a detailed bioavailable Sr isoscape of sub-Saharan Africa, based on extensive direct environmental sampling, the integration of published data, and the use of a machine learning framework. The greatly expanded geographic coverage of our $^{87}Sr/^{86}Sr$ dataset displays a much larger gradient of radiogenic $^{87}Sr/^{86}Sr$ signatures across Africa compared to that suggested in a previous global isoscape[17]. Our study fills considerable gaps in African bioavailable $^{87}Sr/^{86}Sr$ data, specifically for West and West-Central Africa. These regions were key areas of human exploitation and trafficking during the transatlantic slave trade[5] and are home to numerous endangered and trafficked wildlife species today[22]. This high spatial heterogeneity of $^{87}Sr/^{86}Sr$ across Africa has great geolocation potential for provenance applications, particularly in the study of the transatlantic slave trade, but also in wildlife ecology, conservation, and forensics.

Our Sr isoscape contributes more detailed information on the possible origins of enslaved Africans from the Diaspora. In particular, it shows great potential for differentiating the origins of individuals with high $^{87}Sr/^{86}Sr$ ratios (> 0.730), as these highly radiogenic signatures occur only in a few discrete areas of Africa, which coincide with notorious slave trade ports, such as in Angola and southern West Africa (e.g., in southern Côte d'Ivoire and Ghana). We present two prominent archaeological case studies from the African Diaspora involving probable first-generation victims of the slave trade. It is crucial to emphasise that individual histories matter within the transatlantic slave trade. Enslaved people did not comprise a homogeneous living population but were made up of individuals kidnapped from diverse regions of the African continent, with each person possibly representing distinct languages, cultural practices, and traditions that slavery sought to erase[32,39]. Here, we take an individual approach and provide an exemplary demonstration of the potential of our updated Sr isoscape to reconstruct individual African regional origins and, hence, enslaved life histories.

In the case of the two enslaved adult men named *Daba* and *Ganda* from Charleston's Anson Street African Burial Ground with relatively high $^{87}Sr/^{86}Sr$ ratios[10], genetic data associated both of them with West African populations[34]. Within West Africa, our isoscape enabled further narrowing down of their possible origins to discrete

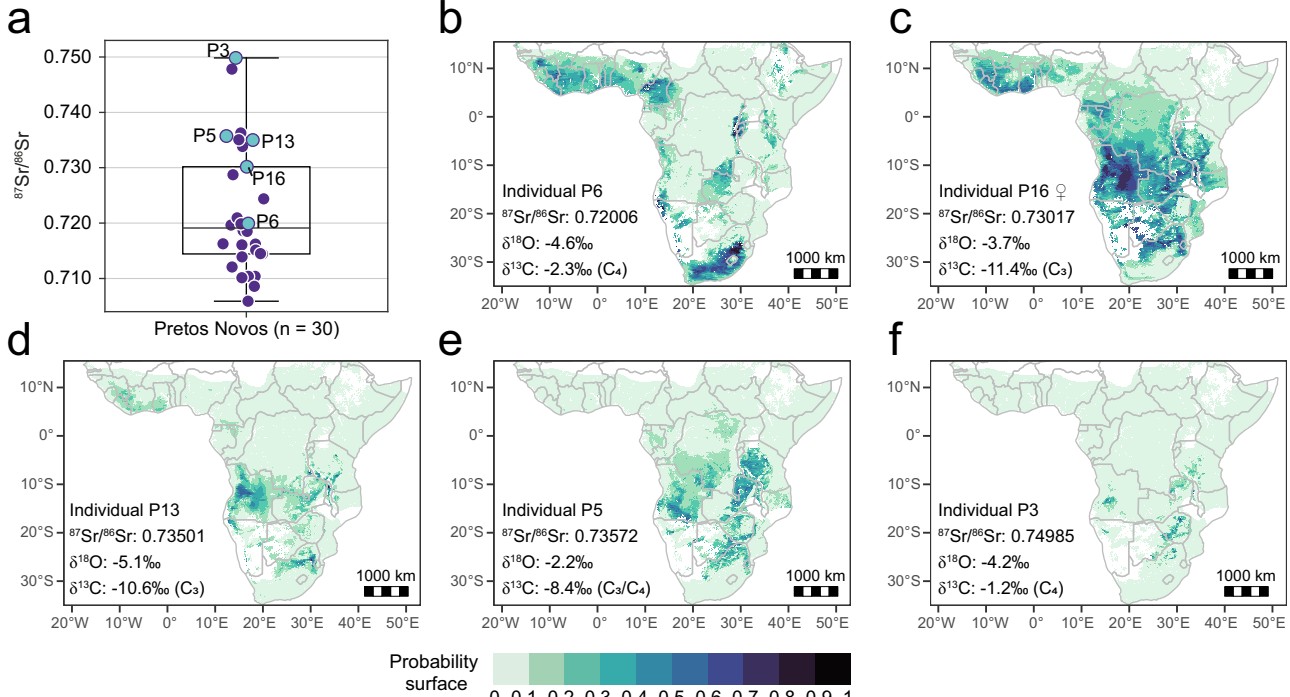

**Fig. 4 | Normalised probability of geographic origins for five enslaved individuals excavated from the Pretos Novos cemetery in Rio de Janeiro (Brazil), using published tooth enamel ⁸⁷Sr/⁸⁶Sr data[9] and our strontium isoscape, in combination with published enamel δ¹⁸O data[35] and the annual mean precipitation oxygen isoscape based on RCWIP data products[84]. a** Boxplot showing the distribution of ⁸⁷Sr/⁸⁶Sr data for 30 individuals buried at the site (Supplementary Data 4), generated using OriginPro 2021 software. The box represents the interquartile range (IQR) from the 25th to the 75th percentile, with a line indicating the median. Whiskers extend to the minimum and maximum values within 1.5 times the IQR from the quartiles, with outliers defined as points beyond this range. We selected five individuals with ⁸⁷Sr/⁸⁶Sr ratios ≥ 0.720, which are very likely not present in the coastal sugar plantation regions of Rio de Janeiro province, suggesting they originated in parts of Africa with more radiogenic bedrock[9]. The isotopic geolocation suggests individual P6 (**b**) may have originated in West or South Africa, whereas the other four individuals (**c–f**) with more radiogenic ⁸⁷Sr/⁸⁶Sr ratios likely came from present-day Angola or the region of South-East Africa. The probability surface was normalised by dividing each cell's value by the maximum probability. Consequently, darker regions represent areas with a higher relative probability of origin compared to other cells, rather than the actual probability values.

regions in southwestern Côte d'Ivoire, southern Ghana, or eastern Guinea (Fig. 3d, f). Interestingly, their likely origins overlap to a considerable extent, meaning that these men could have grown up in the same regions, those historically referred to as the Windward and Gold Coasts. The less radiogenic ⁸⁷Sr/⁸⁶Sr ratio of the adult male *Lima* can be associated with a variety of regions within West Africa. Hence, in his case, the ⁸⁷Sr/⁸⁶Sr data strongly confirms the genetic assessment of origin but does not add much additional information. Two other men, *Kuto* and *Banza*, had been genetically identified to be of West-Central African ancestry[10] and their ⁸⁷Sr/⁸⁶Sr ratios indeed correspond relatively well to the eastern-central parts of Angola, which may indicate they had similar origins within Angola (Fig. 3b, c). The notion of a variety of regional origins for the Anson Street Ancestors is well supported by historic records, which suggest that nearly half of the African captives who arrived in Charleston in the 18ᵗʰ century embarked in West Africa and another third from West-Central Africa[5]. In the case of *Kuto* and *Banza*, the combination of aDNA and isotopic information is particularly fruitful as aDNA data enabled the exclusion of some parts of West Africa. The geographic assignments presented in Fig. 3 may facilitate a deeper understanding of the possible cultural affiliations and ethnic identities of *Daba* and *Ganda* and of *Kuto* and *Banza*.

In the case of the Brazilian slave cemetery, Pretos Novos in Rio de Janeiro, individual P6 may have originated from West or South Africa based on the ⁸⁷Sr/⁸⁶Sr and δ¹⁸O data[35]. This includes various discrete locations within Guinea, central Nigeria, northern Cameroon, as well as broad regions in South Africa (Fig. 4b). The C₄-plant signal in this individual's diet might further exclude the possibility that this person

originated from regions dominated by C₃ crops in West Africa, such as the Rice Coast - the traditional rice-growing region between Guinea and Guinea-Bissau and the western Côte d'Ivoire - as well as the mixed vecegultural zone in most regions of southern West Africa, which were dominated by yams, manioc, and other C₃ root crops[40]. Therefore, within West Africa, the origin of this person can be narrowed down to a very limited area in central Nigeria, within the sorghum-millet zone (C₄ crops), or northern Cameroon, located at the boundary between sorghum-millet and maize-dominated zones (C₄ crops)[40]. While it remains a strong possibility that individual P6 came from South Africa, which has dry farming conditions suitable for C₄ plant growing, historic records suggest a much higher probability of a West African origin, as Brazil received at least 1,540,113 captives from the West African coasts throughout the duration of the slave trade compared to 336,896 captives from South-East Africa and the Indian Ocean[5]. Future aDNA analysis could help distinguish between West and South African origins for individual P6. By contrast, the other four individuals (P3, P5, P13, and P16) show a strong isotopic affiliation with West-Central Africa, particularly Angola (Fig. 4c–f). Each of them is projected to come from different parts of Angola, which has implications for the diversity of languages and traditions these individuals may have brought to the Portuguese colony of Brazil. Indeed, their tooth enamel δ¹³C values suggest that they did not share the same dietary customs in early life. This finding is consistent with historic records indicating that the widespread cultivation of both C₃ (manioc and other C₃ root crops) and C₄ (maize and millet) crops by tribes across Atlantic Central Africa (including Angola), some of which had been introduced by European colonisers as key staples for local populations[41]. These four individuals

also match the isotopic conditions of South-East Africa, particularly northern South Africa and Zambia (Fig. 4c–f). However, for two of them (P16 and P5), the geographic assignment includes areas where our isoscape may not provide reliable predictions (see Fig. 2c). Further, historical records suggest that a South-East African origin for enslaved people in Brazil was less common. The south-eastern coast of Brazil received more than 2.2 million captive Africans from West-Central Africa, compared to 28,000 captives from South-East Africa[5]. Approximately one million captives arrived in Rio de Janeiro between 1765 and 1830 CE alone, primarily arriving from the Angolan ports of Luanda and Benguela as well as St Helena[5]. Therefore, the Angolan origin of these four individuals appears to align better with slave trade records.

These examples from two prominent case studies provide supporting evidence for the diverse origins of first-generation victims of the transatlantic slave trade. Overall, our study demonstrates the potential of using the Sr isoscape of sub-Saharan Africa in combination with other bioarchaeological information (e.g., aDNA and oxygen isotopes) and historical evidence in assessing the life histories of enslaved individuals in the African Diaspora.

## Limitations and recommendations for future sampling
While we are confident in the broad applicability of this sub-Saharan Sr isoscape, its local accuracy and resolution remain closely linked to the density of sample coverage in a given region. We employed a conservative approach by identifying and excluding predictor space from our isoscape that would have required extrapolation. Our isoscape and the environmental distance map transparently depict clear data gaps and regions environmentally dissimilar to the current sample distribution, which would benefit from on-the-ground sample collection. We encourage researchers working in these areas to engage in sample collection to help fill these data gaps through close collaboration with African and international scholars conducting fieldwork in often hard-to-access regions. In addition, we recommend improving the quality and resolution of the environmental variables, alongside sample coverage, that predict $^{87}Sr/^{86}Sr$ variation for sub-Saharan Africa via on-the-ground surveys to enhance the accuracy of isoscape models and their versatile use in the future. This study utilised predicted global environmental data, which are often inaccurate in many parts of the Global South[17].

These limitations in data coverage have only minor implications for the study of the transatlantic slave trade. The West and West-Central African regions most notoriously exploited by human traffickers are well represented in our isoscape and have comparatively low predicted standard error (Fig. 2b). The most considerable gaps in the isoscape are in the Sahel, parts of coastal and inland Namibia, and large parts of Mozambique. While Namibia and the Sahel, with their low population densities, were probably not the focus of slave traders[42], the region of Mozambique and its adjacent hinterlands were heavily impacted during the slave trade of the early to mid-19$^{th}$ century, with at least half a million enslaved people taken from the South-East African region[5]. Unfortunately, our isoscape will be limited in its ability to identify first-generation victims of slavery taken from present-day Mozambique and its adjacent hinterlands until this specific data gap is closed.

## Further applications of the strontium isoscape
Beyond the applications in the archaeology of the transatlantic slave trade, our strontium isoscape, integrated with other isotopic systems (e.g., oxygen, sulphur, hydrogen, and carbon), holds transformative potential for future provenance studies across multiple disciplines. In wildlife conservation, it serves as a powerful tool for identifying the origins of traded wildlife, such as illegally logged timber[43] and endangered species[44] (e.g., elephants[25,45,46] and chimpanzees[47]), aiding efforts to combat poaching and wildlife trafficking by enabling law enforcement to pinpoint hotspots of illegal activity. Furthermore, it supports the study of wildlife dispersal (e.g., chimpanzees[48]) and the mobility of migratory species (e.g., bird and insect species[49]) at large spatial scales, and provides insights into the ecology and adaptations of extinct animal species[50] through the analysis of field, museum, and archaeological specimens. In forensic science, our isoscape is crucial for tracing the geographic origins of unidentified human remains, particularly in the identification of deceased African migrants. For instance, in the 2001 London murder case involving the unidentified child Adam, our study suggests a broader range of origins within Africa than previously assumed[51,52]. In addition, $^{87}Sr/^{86}Sr$ and complementary isotope systems could provide a rapid and comparatively cost-effective approach to help identify the thousands of African migrants who perish in the Mediterranean Sea during their passage to southern Europe. Cattaneo and her colleagues[53] called this the largest humanitarian disaster in Europe since the Second World War, emphasising the rights of the victims to be identified and repatriated if possible. Overall, these diverse applications highlight the usefulness of our isoscape, paving the way for rapid advances in provenance applications in African archaeology, ecology, forensic science, and palaeoecology.

## Methods
### Sample collection
Our $^{87}Sr/^{86}Sr$ dataset comprises a comprehensive sampling of wild plants, soils, and modern/archaeological microfauna (i.e., bones, teeth, and snail shells) conducted over the past decade from mostly remote areas of sub-Saharan Africa ($n = 778$), in combination with previously published $^{87}Sr/^{86}Sr$ data ($n = 1488$) (Supplementary Data 1), spanning 35 African countries in total. Our measured $^{87}Sr/^{86}Sr$ data cover over 24 countries, including 16 countries in which $^{87}Sr/^{86}Sr$ ratios had not been previously reported, primarily in West and West-Central Africa. The majority of samples were obtained within the scope of the Pan African Programme: The Cultured Chimpanzee[54], which involved 38 remote field sites and two nationwide surveys (Liberia[55] and Equatorial Guinea[56]). Other samples (archaeological microfauna/soils) were collected through archaeological fieldwork. All samples (~1–2 g) were collected in locations avoiding potential anthropogenic Sr sources such as farming fields and roads (fertilisers, pesticides, and traffic pollutants) by at least 500 metres. The samples were stored dry (with silica desiccant) and shipped to the University of California Santa Cruz, Arizona State University, and the Max Planck Institute for Evolutionary Anthropology, with all required export and import permits.

We provide information on research permissions and export permits for all collected plant, soil and faunal samples, if required by local regulations, from the following issuing institutions: the Ministère de la Culture de la' Alphabétisation, de l'Artisanat et du Tourisme, Benin (#052/SA/SPMAE/DPC/SGM/DC/MCAAT and #333/SA/SPMAE/DPC/SGM/DC/MCAAT), the Université d'Abomey-Calavi, Benin (#007/2012/UAC/FLASH/DHA/CD/CA), the Ministry of Agriculture, Liberia (#RL-NQES-04302014), Senegal (17/016MEL/DSV/SVPA, research permit #01316/DEF/DGF, export permit #$00079, and export certificate #17/056/MEL/DSV/SVPA), the Nigeria National Park Services, Nigeria (#NPH/GEN/378/V/504), the Rwanda Development Board, Rwanda (export permit #14/RDB-T&C/V.U/14, 10/RDB-T&C/V.U/16, and 11/RDB-T&C/V.U/16), the Direction Générale de la Recherche Scientifique et de l'Innovation Technologique, Côte d'Ivoire (research permit #219/MERS/DGRSIT/TM, #168/MESRS/DGRSIT/mo, and 067/MESRS/DGRI/DR), the Ministére de l'Environnement et de l'Assainissement et du Developpement Durable, Mali (#0243/MEAAD), the Ministère de l'Agriculture de l'Elevage et des Eaux et Forets, Guinea (export permit #0000241, code GN), the Uganda Wildlife Authority, Uganda (export permit #002857, #002858, #002859, and #002860), the Department of Livestock Heath and Entomology, Uganda (#00027786 and #00078131), the Uganda National Council for Science and Technology,

Uganda (#NS 425), the Ministére de la Communication, de la Culture, des Arts et du Tourisme, Burkina Faso (export permit #22/166/MCCAT/SG/DGPC), the Ministry of Livestock, Fisheries and Animal Industries, Cameroon (#AA7384772/COINEPIA/DREPIAS/DDEPIAO/DAEPIAC), the Ministére de le Recherche Scientifique et de l'Innovation Technologique, Republic of the Congo (export permit #163/MRSIT/IRF/DG/DS), the Ministére de l'Agriculture de l'Elevage et des Eaux et Forets, Guinea (research permit #078/2015/OGUIPAR/MEEF/Ck), the Insituto da Bioversidadae e das Acas Protegidas, Guinea-Bissau (permit signed, no # given), the Tanzania Wildlife Research Institute, Tanzania (research permit #2017-336-NA-2017-341, export permits #TWRI/RS-307/Vol.ii/2005/77, #TWRI/RS-307/Vol.ii/2005/78, #TWRI/RS-307/Vol.ii/2005/79, and #TWRI/RS-307/Vol.ii/2005/80), the Tanzania Mining Act, Tanzania (export permit EP/HAN #00000990), the Ministry of Livestock and Fisheries Development of Tanzania, Tanzania (export permit #0000001224), the Geological Survey of Malawi, Malawi (license #1137, certificate #GSD/ZA1137, and export permit #EP06091), the Department of Scientific Services Department of Conservation, Gorongosa National Park, Mozambique (research permit #PNG/DSCi/C239/2022 and export permit #PNG/DSCi/R318/2023), the Wildlife Division, Ghana (#WD/A.185/Vol.5), the Department of Wildlife and Range Management, Ghana (#FRNR/WRM/Vol.2), the Ghana Museums and Monuments Board, Ghana (permit# GMMB/0136/Vol.12/259), the Ministry of Food and Agriculture, Plant Protection and Regulatory Services Directorate, Ghana (export permit #MOFA/PPRSD 0328750), the National Protected Area Authority and Conservation Trust Fund, Sierra Leone (export permit #NPAA/ED/1805/02), the Commission Scientifique sur les Authorisations de Recherche, Gabon (export permit #AE0004/16/MESRC/CENAREST/CG/CST/CSAR, AE0008/15/MESRC/CENAREST/CG/CST/CSAR), and the Direction Centrale des Mines et de la Géologie, Gabon (export permit #0000108).

### $^{87}$Sr/$^{86}$Sr analyses

**Sample preparation and Sr isotope analysis at the University of California Santa Cruz (UCSC).** A total of 608 environmental samples, including plant materials, soil leachates, and microfaunal remains (bones, teeth, and snail shells), were processed at UCSC. The preparation protocols varied according to sample type and were carried out at the Primate Ecology and Molecular Anthropology Laboratory. The procedures for each sample type are as follows. Approximately 1 g of cleaned, dried plant material was weighed and transferred to ceramic crucibles in a recorded sequence. These crucibles were covered with lids and heated in a muffle furnace at 500–800 °C for 8–12 h. After cooling, about 10 mg of plant ash was used for Sr separation. About 1.5 g of air-dried soil was placed in a Retsch mixer mill and processed for 30 minutes. The resulting soil powder was transferred to polypropylene tubes, where 5 mL of 1 M ammonium nitrate ($NH_4NO_3$) solution was added. The samples were agitated for 24 h using a Loopster digital tube rotator, centrifuged, and the supernatant (approximately 4 mL) was filtered through 0.45 µm filters into Teflon beakers[28]. The filtered samples were dried on a hot plate to prepare for Sr separation. For snail shells and faunal tooth enamel, approximately 10 mg of each sample was cleaned with deionized water, followed by ultrasonic treatment with ultrapure acetone for 10 min to remove surface contaminants. After air drying, the samples were prepared for Sr separation. Bone samples were first sandblasted to remove any visible dirt. Each sample was weighed, and about 30 mg was transferred to a beaker. The samples were then cleaned using ultrapure acetone in an ultrasonic bath for 5 minutes, air-dried, and ashed in a muffle furnace at 800 °C for 12 h.

After sample pretreatment, Sr separation was performed in the clean laboratory facilities of the UCSC W.M. Keck Isotope Laboratory using the ion exchange method[57,58]. Samples were digested in 2 mL of 65% $HNO_3$ at 120 °C for 2 h. The completely dissolved component of the solution was evaporated to dryness and re-dissolved in 1 ml of 3 N $HNO_3$. The strontium was then separated from the matrix using Eichrom Sr-Spec resin (50–100 µm). Samples were then re-dissolved in 2 µL of $TaCl_5$ activator solution, and loaded onto degassed rhenium filaments. $^{87}$Sr/$^{86}$Sr ratios were determined using an IsotopX Phoenix X62 Thermal Ionisation Mass Spectrometer (TIMS), with an $^{88}$Sr/$^{86}$Sr ratio of 8.375209 to correct for mass fractionation. The NIST SRM-987 standard was conducted during measurements and yielded an average of 0.710234 ± 0.000021 (2σ, $n = 43$), in agreement with the SRM-987 standard value of 0.710250[59]. Measurements of procedure blanks in each batch (19 samples) were conducted using an Element XR High-Resolution ICP-MS system at the UCSC Plasma Lab, with the results of Sr concentrations below 250 pg.

**Sample preparation and Sr isotope analysis of South African samples.** In a study led by A.M. Zipkin and colleagues, 155 plant samples were collected from South Africa. The plant collection strategy was explicitly guided by mapped geological boundaries. At the Archaeological Chemistry Laboratory at Arizona State University (ASU), each plant sample was thoroughly washed with deionized water to remove any dust, followed by dry ashing in an acid-cleaned alumina crucible at 500 °C for 12 h. The resulting ash from each plant specimen was thoroughly mixed with a clean, disposable spatula, and a sub-sample of ash was then leached with aqua regia made from trace metal-grade acids in an ultra-low metal PFA digestion vessel to yield a stock solution of total non-silicate Sr. An aliquot of each stock solution was then purified of elements (e.g., Ca, Rb, Ba) that could interfere with Sr isotope measurement using the PrepFast automated cation exchange resin system in the ASU Metals, Environmental, and Terrestrial Analysis Laboratory (METAL). $^{87}$Sr/$^{86}$Sr ratios were measured on a Thermo-Fisher Neptune Multi-Collector Inductively Coupled Plasma Mass Spectrometer (MC-ICP-MS). Results were corrected for isobaric interference, mass fractionation, and instrument drift using routine methods[60]. The result of the SRM-987 standard was 0.710237 ± 0.000043 (2σ, $n = 57$), consistent with the NIST SRM-987 standard value of 0.710250[59].

S.R. Copeland and colleagues analysed 15 plant samples from the Sterkfontein Valley, South Africa. The plant materials were dried and ashed at 500 °C for 8 h in a muffle furnace, and about 30–40 mg of plant ash was analysed for $^{87}$Sr/$^{86}$Sr using the Thermo-Fisher Neptune MC-ICP-MS at the Max Planck Institute for Evolutionary Anthropology[58]. The NIST SRM-987 standard yielded the $^{87}$Sr/$^{86}$Sr ratio of 0.710264 ± 0.000035 (2σ, $n = 66$), in agreement with the published standard value[59].

### Isoscape modelling
We used a random forest (RF) algorithm[61] as a generic framework for predictive modelling of spatial and spatio-temporal variables to model the distribution of $^{87}$Sr/$^{86}$Sr ratios across sub-Saharan Africa using our measured $^{87}$Sr/$^{86}$Sr data in combination with published data extracted from the literature ($n = 2266$; see all references in Supplementary Data 1). RF is a modelling approach that calculates and assembles multiple decision trees to predict a response[62], specifically $^{87}$Sr/$^{86}$Sr ratios in this study. Each tree is constructed through a series of steps, splitting the dataset into smaller sets with the smallest intra-variation. Each split is governed by a number of randomly selected features (here selected from a mix of geological, climatic, and topographic predictors). The process is repeated until each leaf has a user-defined value. The prediction error is calculated by cross-validation where each partition sequentially serves as a test dataset. Subsequently, all the trees are used to make predictions in new areas. Due to its ability to integrate various environmental variables into a single predictive framework, RF is widely used for developing Sr isoscapes on different scales.

Initially, we chose 31 independent variables among a range of geological, climatic, topographic, and environmental covariates that may influence bioavailable $^{87}$Sr/$^{86}$Sr ratios in Africa[17,18] (Supplementary Data 2). We did not consider the possible impact of nitrogen and phosphorus, as our sampling strategy systematically avoided anthropogenic Sr sources and these elements' concentrations are generally low in Africa compared to other continents[62].

Missing values in the raster predictors (grid cells with $^{87}$Sr/$^{86}$Sr isotope data but lacking predictor values in that cell) were replaced with values from the nearest grid cell for those with categorical data and with bedrock age. For continuous predictors, we replaced the NAs in the dataset with the mean of the 5 nearest neighbours (see associated script infillNN.R). To avoid collinearity among predictors, we reduced this dataset with various methods, including the VSURF algorithm[63], reduction of multicollinearity with Pearson's correlations, variance inflation factors, and visual inspections of predictor correlations. The final set of 11 predictors included dust, terrane age, soil clay content, maximum age of rocks, soil cation exchange capacity, sea salt deposition, mean annual precipitation, elevation, soil organic carbon content, srsrq3 (bedrock model), and lithology. Before modelling, we optimised two model hyperparameters (number of variables randomly sampled at each split and maximum node size). Twenty samples were set aside as a validation set. The RF model was trained on 2246 data points and validated using a k-fold cross-validation method (5 folds, 10 repetitions). We assessed model accuracy using the mean $R^2$ and RMSE, as well as the error in predicting the validation set.

Unlike previous isoscape studies[17,18], we did not apply the median $^{87}$Sr/$^{86}$Sr ratios for data points within the same grid cell (here 1 km$^2$) but instead treated all data as independent. The rationale for using the median is that samples in close proximity (within the same grid cell) have the same predictor values, i.e., landscape overlap[64], and this leads to pseudoreplication and violates the assumption of independence. Firstly, we argue that our sampling is sufficiently independent, that the resampling is caused by low predictor resolution, and that aggregating data leads to other types of error. In fact, our $^{87}$Sr/$^{86}$Sr samples were collected from various organisms and substrates, often hundreds of metres apart. Yet, the lack of higher-resolution maps for predictor variables at the continental scale results in these samples originating from the same cell and, thus the same predictor space (or landscape). While data from the same grid cell (1 km$^2$) will share the same environmental properties (overlapping landscapes), their $^{87}$Sr/$^{86}$Sr ratios will differ due to natural variation of $^{87}$Sr/$^{86}$Sr between sample locations within that cell, as well as among different sample types (e.g., various plants, soils, and animals), which are influenced by various factors affecting Sr sourcing[17,36]. In addition, aggregating data using medians would drastically reduce the effective sample size, leading to increased Type I error[64]. Treating each data point as independent allows us to integrate the natural variation in $^{87}$Sr/$^{86}$Sr ratios within each grid cell as an important part of the algorithm training process. Secondly, we argue that the independence assumption refers to the model residuals (the part of the variance not explained by the model) rather than the sampling[65,66]. Zuckerberg et al.[66] demonstrated that reducing landscape oversampling does not necessarily reduce spatial autocorrelation (an indication of residual dependence in spatial models). Spatial autocorrelation in the residuals would indicate that there is an effect of sample proximity effects or landscape overlap that is not explained by the model. Therefore, to determine if our approach violates statistical independence, we assessed potential spatial autocorrelation in the model residuals using a cross-correlogram[67].

To create the isoscape, we used the fitted model to predict $^{87}$Sr/$^{86}$Sr ratios in unsampled areas. We also estimated standard errors based on out-of-bag predictions using the infinitesimal jackknife method for bagging[68]. Although RF models have been proven efficient for spatial prediction, they do not reliably extrapolate into areas that differ significantly from the sampled conditions[69]. We employed the mobility-oriented parity metric (MOP[70]) to identify areas of strict extrapolation (where conditions fall outside of the sampled range) and to calculate potential combinational extrapolation areas (where conditions are within the range but their combinations may be distinctive). All grid cells where at least one predictor's values were outside the sampled range were removed from the isoscape. To measure the environmental similarity of each grid cell (a measure of combinational extrapolation), we calculated the multivariate Mahalanobis distance to the nearest 10% of the sampled cloud.

All analyses were performed in the R environment[71] using various packages for specific analyses, including data handling and visualisation (tidyverse v. 1.3.1[72]), spatial data handling (sf v. 1.0–6[73]; terra v. 1.5–17[74]; rnaturalearth v. 0.1.0[75]), variable selection (VSURF v. 1.1.0[63]; spatialRF v. 1.1.3[76]), random forest regressions (ranger v. 0.13.1[77]; caret v. 6.0-92[78]; tuneRanger v. 0.5[79]), mobility-oriented parity (mop v. 0.1.1[80]), spline correlograms (ncf v. 1.3-2[81]), partial dependence plots (pdp v. 0.7.0[82]), and to infer geographic origin (assignR v. 2.1.1[83]).

## Assignment of human $^{87}$Sr/$^{86}$Sr ratios

We estimated the probable geographic origins of ten enslaved Africans from two slave cemeteries in the African Diaspora. To achieve this, we applied a predicted strontium isoscape of sub-Saharan Africa (mean values) and its estimated standard error (calculated using the infinitesimal jackknife) resulting from the random forest regression and the continuous-surface assignment framework from the R package assignR using the pdRaster function[83]. The resulting probability surfaces for each sample were normalised for comparison (each cell's probability was divided by the maximum probability).

The human enamel $^{87}$Sr/$^{86}$Sr data were sourced from previous studies, including five individuals from the Anson Street African Burial Ground (1760–1790 CE) in Charleston, USA[10] and five individuals from the Pretos Novos cemetery (1769–1830 CE) in Rio de Janeiro, Brazil[9]. From both sites, we selected individuals with comparatively radiogenic $^{87}$Sr/$^{86}$Sr ratios exceeding 0.720, as such ratios are not found in coastal South Carolina[29] and are unlikely to be present in the coastal sugar plantation regions of Rio de Janeiro province.

For the Anson Street ancestors, we combined $^{87}$Sr/$^{86}$Sr data with aDNA evidence to identify first-generation victims of the slave trade, using published genetic findings to narrow predictions of individual origin to specific African regions (West Africa versus West-Central Africa)[34].

For the ancestors from the Pretos Novos cemetery, we employed dual-isotope ($^{87}$Sr/$^{86}$Sr and $\delta^{18}$O) geographic assignments due to the availability of $\delta^{18}$O data from human tooth enamel[35]. Here, we additionally used the modern precipitation oxygen isoscape of Africa[84] in the assigned R framework. We first adjusted human tooth carbonate $\delta^{18}$O values to those of drinking water using established equations. Specifically, we converted enamel carbonate $\delta^{18}$O values from V-PDB to V-SMOW using the equation from Coplen et al.[85] ($\delta^{18}O_{carb-VSMOW} = 1.03091 \times \delta^{18}O_{carb-VSMOW} + 30.91$), then converted carbonate $\delta^{18}$O values to phosphate $\delta^{18}$O values using the equation from Iacumin et al.[86] ($\delta^{18}O_{p-VSMOW} = 0.98 \times \delta^{18}O_{carb-VSMOW} - 8.5$), and finally applied the equation from Daux et al.[87] to convert the resulting $\delta^{18}$O values to drinking water $\delta^{18}$O values ($\delta^{18}O_{w-VSMOW} = 1.54 \times \delta^{18}O_{p-VSMOW} - 33.72$). We compared the calculated human drinking water $\delta^{18}$O values with the annual mean precipitation oxygen isoscape of Africa (RCWIP)[84], accounting for the uncertainty of the isoscape and an additional uncertainty of 1‰ (considering the error ranges of the conversions above[88,89]).

## Reporting summary

Further information on research design is available in the Nature Portfolio Reporting Summary linked to this article.

## Data availability

The software packages utilised for the analysis are publicly available and are cited either in the Methods section or in the Supplementary Information. All data (including both original data and previously published data) generated or analysed during this study are included in the main text or supplementary information files. These data are also available in the Figshare repository at: https://doi.org/10.6084/m9.figshare.23118212.v1, which is publicly accessible with no restrictions. Source data for Figs. 1, 2, 3, and 4 can be found in the Supplementary Data file. Source data for Supplementary Figs. 1, 2, 3, 5, and 6 are also included in the Supplementary Data file. All remaining environmental samples are stored at the University of California Santa Cruz (Santa Cruz, CA, USA), and can be accessed upon request by contacting Vicky M. Oelze at voelze@ucsc.edu. Source data are provided in the Source Data file. Source data are provided in this paper.

## Code availability

The R scripts used for data analysis, all the data files for modelling and the resulting isoscapes are available at: https://doi.org/10.6084/m9.figshare.23118212.v1.

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

## Acknowledgements

We dedicate this work to our colleague and mentor, the late Christophe Boesch (1951–2024), who co-founded the Pan African Programme, which was foundational to this project. We thank Terry Blackburn, Gavin Piccione, and Graham Harper Edwards for providing TIMS support in the UCSC W.M. Keck Isotope lab, and Brian Dreyer for his help with the HR-ICP-MS in the UCSC Plasma lab. Numerous individuals and institutions assisted with this project by collecting samples in the field. We thank Stephen Dueppen, Matt Kroot, Sirio Canos-Donnay, Martijn Ter Heegde, Manasseh Eno-Nku, Joshua M. Linder, John Hart, Thurston Cleveland Hicks, Theophile Desarmeaux, Vianet Mihindou, Marcel Ketchen Eyong, Laura Kehoe, Lucy D'Auvergne, Els Ton, Luz Calia Miramontes Sequeiros, Theo Freeman, Emilien Terrade, Lilah Sciaky, Hilde Vanleeuwe, Jean Claude Dengui, Abel Nzeheke, Michael Masozera, Nicolas Ntare, Michael Kaiser, Henk Eshuis, Geoffrey Muhanguzi, Martha Robbins, Alhaji Malikie Siaka, Gwyneth Gordon, Pauline Wiessner, Curtis W. Marean, Ariel Anbar, Marc Stalmans, Greg Carr, the Rangers of Gorongosa National Park, Didier N'Dah, L. Llandu, Constantin Lubini, Musuyu Désiré Muganza, Jean Jacques Muyembe, Bryna Hull, Baba Ceesay, Hassum Ceesay, Lourenço Vaz Rodrigues, Valeria Vdovina, Gino Gomes, Maria da Conceição Freitas, and Mawunu Monizi for their help during fieldwork or their support of our project. In addition, we are grateful to the numerous institutions and governmental organisations for their support, including the Direction du Patrimoine Culturel (Senegal), the Ministère de la Recherche Scientifique et de l'Innovation (Cameroon), the Ministère des Forets et de la Faune (Cameroon), the Ministère des Eaux et Forêts (Côte d'Ivoire), the Ministère de l'Enseignement Supérieur et de la Recherche Scientifique (Côte d'Ivoire), the Institut Congolais pour la Conservation de la Nature (Democratic Republic of the Congo), the Ministère de la Recherche Scientifique (Democratic Republic of the Congo), the Université de Kinshasa and the Departments de Biologie et Environnement (Democratic Republic of the Congo), the Institut National pour l'Etude et la Recherche Agronomiques (Democratic Republic of the Congo), the Institut National de Recherche Biomédicale (Democratic Republic of the Congo), the Agence Nationale des Parcs Nationaux (Gabon), the Centre National de la Recherche Scientifique (Gabon), the Société Equatoriale d'Exploitation Forestière (Gabon), the Department of Wildlife and Range Management (Ghana), the Forestry Commission (Ghana), the Ministère de l'Agriculture de l'Elevage et des Eaux et Forets (Guinea), the Instituto da Biodiversidade e das Áreas Protegidas (Guinea-Bissau), the Ministro da Agricultura e Desenvolvimento Rural (Guinea-Bissau), the Forestry Development Authority (Liberia), the Ministry of Agriculture (Liberia), the Eaux et Forêts (Mali), the Ministre de l'Environnement et de l'Assainissement et du Developpement Durable (Mali), the Conservation Association of Mbe Mountains (Nigeria), the National Park Service (Nigeria), the Ministere de l'Economie Forestiere (Republic of the Congo), the Ministere de le Recherche Scientifique et Technologique (Republic of the Congo), the Agence Congolaise de la Faune et des Aires Protégées (Republic of the Congo), the Ministry of Education (Rwanda), the Rwanda Development Board (Rwanda), the Direction des Eaux, Forêts, Chasses et de la Conservation des Sols (Senegal), the Ministry of Agriculture, Forestry and Food Security (Sierra Leone), the National Protected area Authority (Sierra Leone), the Tanzania Commission for Science and Technology (Tanzania), the Tanzania Wildlife Research Institute (Tanzania), the Makerere University Biological Field Station (Uganda), the Uganda National Council for Science and Technology (Uganda), the Uganda Wildlife Authority (Uganda), the National Forestry Authority (Uganda), the National Institute for Forestry Development and Protected Area Management (Equatorial Guinea), the Ministry of Agriculture and Forests (Equatorial Guinea), the Ministry of Fisheries and Environment (Equatorial Guinea), the Ministere de la culture de la' Alphabétisation, de l'Artisanat et du Tourisme (Benin), the Universite d'Abomey-Calavi (Benin), the Gorongosa National Park (Mozambique), the Acção para o Desenvolvimento (Guinea-Bissau), the Korup Rainforest Conservation Society (Cameroon), the WWF (Campo Ma'an NP, Cameroon), the Ebo Forest Research Station (Cameroon), the Project Grands Singes, the La Belgique (Cameroon), the Tai Chimpanzee Project (Côte d'Ivoire), the Wild Chimpanzee Foundation (Côte d'Ivoire), the Lukuru Wildlife Research Foundation (Democratic Republic of the Congo), the WCS Albertine Rift Programme (Democratic Republic of the Congo), the WWF Congo Basin (Democratic Republic of the Congo), the Loango Ape Project (Gabon), the Aspinall Foundation (Gabon), the Station d'Etudes des Gorilles et Chimpanzés (Gabon), the Kwame Nkrumah University of Science and Technology (Ghana), the Wild Chimpanzee Foundation (Guinea), the Foundation Chimbo (Guinea-Bissau), the Acção para o Desenvolvimento, the Wild Chimpanzee Foundation (Liberia), the Gashaka Primate Project (Nigeria), the Wildlife Conservation Society (Nigeria), the WCS (Conkouati-Douli NP, Republic of the Congo), the Goualougo Triangle Ape Project (Republic of the Congo), the Nouabalé-Ndoki Foundation (Republic of the Congo), the Gishwati Chimpanzee Project (Rwanda), the Nyungwe-Kibira Landscape (Rwanda-Burundi), the Fongoli Savanna Chimpanzee Project (Senegal), the Field assistants and volunteers from the Jane Goodall and Institute Spain and Senegal, the Greater Mahale Ecosystem Research and Conservation (Tanzania), the Budongo Conservation Field Station (Uganda), and the Ngogo Chimpanzee Project (Uganda). Financial support for this research was provided by the Webster Foundation (to V.M.O.) and the University of California Santa Cruz (to V.M.O.). Further funding for sample collection was provided by the Max Planck Society (to C.B. and H.S.K.), the Max Planck Society Innovation Fund (to C.B. and H.S.K.), and the Heinz L. Krekeler Foundation (to C.B., H.S.K., M.A. and V.M.O.).

## Author contributions

All authors contributed extensively to the work presented in this paper. V.M.O. conceptualised this study. G.Bock., M.A., A.A., S.A., F.A., E.A.A., E.B., D.B., M.Bess., R.Bobe., M.Bonn., G.Braz., S.B., K.C.L., S.C., R.C., C.Cipo., H.C., S.R.C., K.C., A.M.C., C.Coup., B.C., D.J.d.R., T.D., P.D., K.D., E.D., D.D., A.D., V.E.E., M.F., B.F., L.G., Y.G.Y., A.G., C.G., R.G.C., A.H.G., A.-C.G., V.G., C.C.G., A.H., D.H., V.H., R.A.H.-A., G.H., I.I., K.J.J., S.J., J.J., P.K., M.Kambe., M.Kambi., I.K., K.J.K., K.E.L., V.Lape., J.L., B.Lars., T.Laut., P.l.R., V.Lein., M.L., A.L., T. Lüde., G.M., S.M., R.M., P.J.M., A.C.M., P.M., J.C.M., D.M., F.M., M.M., E.Neil, S.N., P.N., E.Norm., L.J.O., O.D., L.P., A.P., J.P., S.R., F.G.R., M.P.R., A.R., C.S., V.S., M.S., T.E.S., F.A.S., N.T., L.R.T., A.T., L.T., J.v.S., V.V., N.W.N., E.G.W., J.W., R.M.W., K.Y., A.M.Z., K.Z., H.S.K., C.B. and V.M.O. contributed to sample collection in the field. X.W., V.M.O., R.Bouc., B.Lowr., S.R.C. and A.M.Z. performed laboratory work. G.Bock. and X.W. analysed the data and conducted the modelling. X.W., V.M.O. and G.Bock. wrote the manuscript, with feedback from all authors.

## Competing interests

The authors declare no competing interest.

## Additional information

Xueye Wang[1,2], Gaëlle Bocksberger [2,3], Mimi Arandjelovic [4], Anthony Agbor[4], Samuel Angedakin[4], Floris Aubert[5], Emmanuel Ayuk Ayimisin[4], Emma Bailey[4], Donatienne Barubiyo[4], Mattia Bessone [4], René Bobe [6,7], Matthieu Bonnet[4], Renée Boucher[2], Gregory Brazzola[4], Simon Brewer [8], Kevin C. Lee[4], Susana Carvalho [6,7], Rebecca Chancellor[9], Chloe Cipoletta[10], Heather Cohen [4], Sandi R. Copeland[11], Katherine Corogenes[4], Ana Maria Costa [12], Charlotte Coupland[4], Bryan Curran[4], Darryl J. de Ruiter[13], Tobias Deschner[14], Paula Dieguez [15], Karsten Dierks[4], Emmanuel Dilambaka[10], Dervla Dowd[5], Andrew Dunn[10], Villard Ebot Egbe[4], Manfred Finckh [16], Barbara Fruth [17], Liza Gijanto[18], Yisa Ginath Yuh [4], Annemarie Goedmakers [19], Cameron Gokee[20], Rui Gomes Coelho [21,22], Alan H. Goodman[23], Anne-Céline Granjon [4], Vaughan Grimes[24], Cyril C. Grueter [25], Anne Haour [26], Daniela Hedwig[27], Veerle Hermans [28], R. Adriana Hernandez-Aguilar [29,30], Gottfried Hohmann[4], Inaoyom Imong[10], Kathryn J. Jeffery [31], Sorrel Jones [32], Jessica Junker[4], Parag Kadam [33], Mbangi Kambere[4], Mohamed Kambi[4], Ivonne Kienast[4], Kelly J. Knudson[34], Kevin E. Langergraber[34], Vincent Lapeyre[5], Juan Lapuente[4], Bradley Larson[4], Thea Lautenschläger[35,36], Petrus le Roux [37], Vera Leinert[5], Manuel Llana[30], Amanda Logan[38], Brynn Lowry[2], Tina Lüdecke [39], Giovanna Maretti[4], Sergio Marrocoli[4], Rumen Fernandez[4], Patricia J. McNeill[40], Amelia C. Meier [41], Paulina Meller[16], J. Cameron Monroe[2], David Morgan[42], Felix Mulindahabi[10], Mizuki Murai[4], Emily Neil [4,43], Sonia Nicholl [4], Protais Niyigaba[10], Emmanuelle Normand[44], Lucy Jayne Ormsby[4], Orume Diotoh[45], Liliana Pacheco [46], Alex Piel[4,47], Jodie Preece[4], Sebastien Regnaut [5], Francois G. Richard[48], Michael P. Richards [49], Aaron Rundus[9], Crickette Sanz [50,51], Volker Sommer[47,52], Matt Sponheimer[53], Teresa E. Steele [40], Fiona A. Stewart[4,47], Nikki Tagg [28], Luc Roscelin Tédonzong [28], Alexander Tickle[4], Lassané Toubga [54], Joost van Schijndel[4], Virginie Vergnes[44], Nadege Wangue Njomen[55], Erin G. Wessling[56,57], Jacob Willie [28], Roman M. Wittig[58,59], Kyle Yurkiw[4], Andrew M. Zipkin[34], Klaus Zuberbühler[60], Hjalmar S. Kühl[4,15,61,62], Christophe Boesch [4,63] & Vicky M. Oelze [2] ✉

[1]Center for Archaeological Science, Sichuan University, Chengdu, China. [2]Anthropology Department, University of California Santa Cruz, Santa Cruz, CA, USA. [3]Senckenberg Biodiversity and Climate Research Centre (SBiK-F), Frankfurt, Germany. [4]Max Planck Institute for Evolutionary Anthropology, Leipzig, Germany. [5]Wild Chimpanzee Foundation, Leipzig, Germany. [6]Institute of Human Sciences, School of Anthropology, University of Oxford, Oxford, UK. [7]Gorongosa National Park, Sofala, Mozambique. [8]Department of Geography, University of Utah, Salt Lake City, UT, USA. [9]Departments of Anthropology and Sociology, West Chester University, West Chester, PA, USA. [10]Wildlife Conservation Society, Bronx, NY, USA. [11]Environmental Stewardship Group, Los Alamos National Laboratory, Los Alamos, NM, USA. [12]Laboratório de Arqueociências (LARC)-DGPC and CIBIO | BIOPOLIS, 1300-418, Lisbon/ IDL - Instituto Dom Luiz, University of Lisbon, Lisbon, Portugal. [13]Department of Anthropology, Texas A&M University, College Station, TX, USA. [14]Comparative BioCognition, Institute of Cognitive Science, University of Osnabrück, Osnabrück, Germany. [15]German Centre for Integrative Biodiversity Research (iDiv), Leipzig, Germany. [16]Ecological Modeling, Institute of Plant Science and Microbiology, University of Hamburg, Hamburg, Germany. [17]Max Planck Institute of Animal Behavior, Konstanz, Germany. [18]Department of Anthropology, St. Mary's College of Maryland, St. Mary's City, MD, USA. [19]Chimbo Foundation, Oudemirdum, the Netherlands. [20]Department of Anthropology, Appalachian State University, Boone, NC, USA. [21]Department of Archaeology, Durham University, Durham, UK. [22]Centre for Archaeology, University of Lisbon, Lisbon, Portugal. [23]School of Natural Science, Hampshire College, Amherst, MA, USA. [24]Department of Archaeology, Memorial University, St. John's, Newfoundland, Canada. [25]School of Human Sciences, The University of Western Australia, Crawley, Western Australia, Australia. [26]Sainsbury Research Unit for the Arts of Africa, Oceania and the Americas, University of East Anglia, Norwich, United Kingdom. [27]K. Lisa Yang Center for Conservation Bioacoustics, Cornell Lab of Ornithology, Cornell University, Ithaca, NY, USA. [28]Centre for Research and Conservation, Royal Zoological Society of Antwerp, Antwerp, Belgium. [29]Department of Social Psychology and Quantitative Psychology, University of Barcelona, Serra Hunter Programme, Barcelona, Spain. [30]Jane Goodall Institute Spain and Senegal, Dindefelo Biological Station, Dindefelo, Kedougou, Senegal. [31]School of Natural Sciences, University of Stirling, Stirling, Scotland, UK. [32]RSPB Centre for Conservation Science, Royal Society for the Protection of Birds, The David Attenborough Building, Cambridge, UK. [33]Warnell School of Forestry and Natural Resources, University of Georgia, Athens, GA, USA. [34]School of Human Evolution

and Social Change, Arizona State University, Tempe, AZ, USA. [35]Department of Biology, Institute of Botany, Faculty of Science, Technische Universität Dresden, Dresden, Germany. [36]Botanical Garden, Universität Hamburg, Hamburg, Germany. [37]Department of Geological Sciences, University of Cape Town, Rondebosch, South Africa. [38]Department of Anthropology, Northwestern University, Evanston, IL, USA. [39]Emmy Noether Group for Hominin Meat Consumption, Max Planck Institute for Chemistry, Mainz, Germany. [40]Department of Anthropology, University of California at Davis, Davis, CA, USA. [41]Hawai'i Institute of Marine Biology, University of Hawai'i at Manoa, Kane'ohe, HI, USA. [42]Lester E. Fisher Center for the Study and Conservation of Apes, Lincoln Park Zoo, Chicago, IL, USA. [43]School of Geography and the Environment, University of Oxford, Oxford, UK. [44]Wild Chimpanzee Foundation (WCF), Abidjan, Côte d'Ivoire. [45]Korup Rainforest Conservation Society, Korup National Park, Mundemba, SW Region, Cameroon. [46]Save the Dogs and Other Animals, Cernavoda, Cernavoda, CT, Romania. [47]Department of Anthropology, University College London, London, UK. [48]Department of Anthropology, University of Chicago, Chicago, IL, USA. [49]Department of Archaeology, Simon Fraser University, Burnaby, Canada. [50]Department of Anthropology, Washington University in St. Louis, St. Louis, MO, USA. [51]Wildlife Conservation Society, Congo Program, Brazzaville, Republic of the Congo. [52]Gashaka Primate Project, Serti, Taraba, Nigeria. [53]Department of Anthropology, University of Colorado Boulder, Boulder, CO, USA. [54]Université Joseph KI-ZERBO, Ouagadougou, Burkina Faso. [55]WWF Cameroon Country Office, Yaoundé, Cameroon. [56]Department of Human Evolutionary Biology, Harvard University, Cambridge, MA, USA. [57]University of St. Andrews, St. Andrews, Scotland, UK. [58]Institut des Sciences Cognitives, University of Lyon 1, Bron, France. [59]Tai Chimpanzee Project, Centre Suisse de Recherches Scientifiques, Abidjan, Côte d'Ivoire. [60]Université de Neuchâtel, Institut de Biologie, Neuchâtel, Switzerland. [61]Senckenberg Museum for Natural History Görlitz, Senckenberg-Member of the Leibniz Association, Görlitz, Germany. [62]International Institute Zittau, Technische Universität Dresden, Zittau, Germany. [63]Deceased: Christophe Boesch. ✉e-mail: voelze@ucsc.edu

