## [Peer Review file · Nature Communications]

Strontium isoscape of sub-Saharan Africa allows tracing origins of victims of the transatlantic slave trade

Corresponding Author: Dr Vicky Oelze

Version 1:

Reviewer comments:

Reviewer #1

(Remarks to the Author)

Overall, I believe the authors have very carefully taken into account all my reviews and I appreciate the time and efforts they have put into improving this manuscript. I believe this manuscript is very improved and is a very interesting use of strontium isotopes that will certainly be highly cited.

I will first reiterate why I think this study deserve publication in a high-impact journal like Nature Communications.

- 1) This study is data rich with close to ~800 new bioavailable strontium isotope data from very remote area in Africa a region in which there are many interests to apply geolocation but where data were very scarce
- 2) The pilot strontium isoscape for Africa has broad reaching relevance to many fields. Those include archeologists, anthropologist, forensic specialists but also ecologists, paleoecologists and environmental scientists. So I believe the product will be broadly used and will be of great interest to Nat Comm readers
- 3) This study provides some advances to model strontium isotopes. This is a booming field with a growing community and the study provides some new interesting tools to assess uncertainty of the isoscape and support improvement in sampling strategy to fill data gap.
- 4) The rewriting of the paper towards African slaves identification is great as well as the rewriting of the discussion combining isotopes and genomics. I think it creates a more focused story that has ethical and historical relevance for humanity that will again be of interest to Nat. Comm broad reader base.

So I would argue that this study should be published quickly as the community needs it and it has already been through very significant revisions. Before it is published I have some minor comments:

- 1) The authors need to add a little bit more details on the geographic assignments. I have not had the time to look at the script this time but it needs to be clearer what function in assignR was used with what inputs. For the strontium isoscape, are talking about the predicted mean and pseudo-sd from the quantile random forest regression? Correct? If yes it needs to be specified. For the oxygen, the mean precipitation d18O values and its associated uncertainty +1 per mile? Correct? Please detail this a bit more, what function was used and with what tuning.
 - 2) I am not fully convinced about their reply on pseudoreplication. I totally agree with the nice new paragraph about how local uncertainty associated with the range of bioavailable $87\text{Sr}/86\text{Sr}$ values is meaningful and needs to be incorporated into the modeling one way or another. However, if you take multiple points per grid cell it creates pseudo-replication by definition even if the values are slightly different. I don't have a clear solution to this problem but I did not fully understand their reply. So I am not asking the author to redo any modeling but I would like to understand why Zuckerberg, B. et al. suggest that resampling at one site does not involve pseudoreplication as this is the study cited to justify this statement? And how this cross-correlogram from Bjørnstad, O. N. & Falck, W. help us to see that there is no spatial-autocorrelation linked to pseudo-replication? Could you clarify those two points with a bit more details in the text?
 - 3) I think the author could combine the last two paragraphs (applications for wildlife/application for forensics) into one concluding paragraph describing the broader implications/applications of their isoscape. I find the last two paragraphs still distract a bit from the main message on African slave origin as they are pretty long. So I would simply have one final paragraph where the authors describe briefly all the cool applications and potential solutions that this new isoscape brings to African ecology, forensics, paleoecology... Short and exciting with a few refs.
- Line 173: referred to what? Weird sentence ending?

(Remarks on code availability)

I had reviewed and commented the code last time in great details. The authors have taken into account all my comments in their reviews. I did not have time to rerun fully the script this time but I believe most of the issues have been addressed.

Reviewer #2

(Remarks to the Author)

This version of the manuscript addresses most of the problems I identified previously. My major concern about this version is to do with the interpretation of Figs 3 and 4 (see below). I also have a number of relatively minor comments, listed below.

First paragraph: I am not at all sure that the slave trade was “the most notable migration event” in Africa – the Bantu expansion was arguably as influential. Add “forced” before “migration”, as you say a couple of lines on, and no-one would argue.

It is untrue that “little to nothing is known about what regions and communities they [African slaves] had been taken from” – there is a large body of scholarship on this in African history.

Second paragraph: “ $^{87}\text{Sr}/^{86}\text{Sr}$ ratios in animal and human body tissues primarily mirror biologically available $^{87}\text{Sr}/^{86}\text{Sr}$ incorporated via the consumption of plants and water from the local substrate with minimal fractionation which can be corrected during mass spectrometric analysis (Supplementary Note S1)” is still incorrect. This fraction IS corrected for (not “can be”), not during mass spectrometric analysis, rather during post-measurement data processing. These might seem like picky points, but science requires precise use of language.

Second paragraph of Results: Specify the “several soil properties” used as predictors in your isoscape.

It is untrue that “that RF models are unable to extrapolate into areas without matching training data”. What is true is that we can have little confidence in such extrapolations.

The level of “match” between $^{87}\text{Sr}/^{86}\text{Sr}$ of the skeletons and the areas they may have come from is quite uneven. Fig 3e shows very good agreement, but 3d and 3f appear to offer at best 50-60% match. Similarly, in Fig 4, 4c offers quite good agreement but 4f does not. This requires some discussion.

Does the DNA from Daba, Lima and Ganda specifically link them to the southern coast of West Africa, or just to West Africa in general? There were major West African slaving ports that fall outside the limits of this isoscape, especially Goree Island off Dakar. Could they have come from there?

In Fig 4, what about the $\delta^{13}\text{C}$ values? Do these help narrow down the possibilities?

Note that St Helena was simply a pick-up point – slaves taken on board there could have come from many areas. The same is true of Mozambique- slaves were brought from the interior, likely spanning several contemporary countries, and shipped from Mozambique. Individuals recorded as “of Mozambique” did therefore not necessarily originate from what we call Mozambique today.

Under “Limitations and future work”, the phrase “regions with strict data extrapolation” needs re-wording, making it clear that these are regions lacking actual analyses.

(Remarks on code availability)

Reviewer #4

(Remarks to the Author)

(Remarks on code availability)

Version 2:

Reviewer comments:

Reviewer #1

(Remarks to the Author)

The authors have addressed all my comments.

1) I appreciate the detailed answer regarding pseudoreplication and the cross-correlogram.

2) I think the methods is improved with the new additions. I do think that the oxygen isoscape uncertainty could have been generated potentially in a better way by compiling oxygen data in enamel and then using calRaster() to generate a true uncertainty layer. However, I believe the author approach with a 1 per mile uncertainty addition to the precipitation layer is conservative.

3) The implications section of the manuscript is much improved.

So overall I believe that manuscript should be published.

(Remarks on code availability)

I had reviewed the code in details previously and provided detailed comments about how to make it more reproducible. I believe this has all been addressed.

Reviewer #2

(Remarks to the Author)

I am satisfied that this revision deals with all my earlier questions and criticisms. Two very minor points:

It is inconsistent to say in line 162 that “the $^{87}\text{Sr}/^{86}\text{Sr}$ ratios of a location primarily relate to the underlying bedrock geology” and in lines 245-6 that “dust deposition is the most important factor influencing spatial bioavailable $^{87}\text{Sr}/^{86}\text{Sr}$ variation”

Line 422: “board” should be “broad”

Once the authors have dealt with these, I believe the manuscript should be published.

(Remarks on code availability)

Point-by-Point Response to Reviewers – Author response in blue font throughout

(please note: Line numbers refer to the numbering in the clean version of the manuscript)

Reviewers' comments:

Reviewer #1 (Remarks to the Author):

Comment 1:

Overall, I believe the authors have very carefully taken into account all my reviews and I appreciate the time and efforts they have put into improving this manuscript. I believe this manuscript is very improved and is a very interesting use of strontium isotopes that will certainly be highly cited.

I will first reiterate why I think this study deserve publication in a high-impact journal like Nature Communications.

1) This study is data rich with close to ~800 new bioavailable strontium isotope data from very remote area in Africa a region in which there are many interests to apply geolocation but where data were very scarce

2) The pilot strontium isoscape for Africa has broad reaching relevance to many fields. Those include archeologists, anthropologist, forensic specialists but also ecologists, paleoecologists and environmental scientists. So I believe the product will be broadly used and will be of great interest to Nat Comm readers

3) This study provides some advances to model strontium isotopes. This is a booming field with a growing community and the study provides some new interesting tools to assess uncertainty of the isoscape and support improvement in sampling strategy to fill data gap.

4) The rewriting of the paper towards African slaves identification is great as well as the rewriting of the discussion combining isotopes and genomics. I think it creates a more focused story that has ethical and historical relevance for humanity that will again be of interest to Nat. Comm broad reader base.

So I would argue that this study should be published quickly as the community needs it and it has already been through very significant revisions. Before it is published I have some minor comments:

Response: Thank you for your thorough and positive feedback. We are delighted that you find our study to be of high significance and potential impact. We greatly appreciate your detailed endorsement of our work and the acknowledgment of our efforts in addressing your previous reviews. Based on your minor comments, please find our detailed responses and corresponding revisions below.

Comment 2:

1) The authors need to add a little bit more details on the geographic assignments. I have not had the time to look at the script this time but it needs to be clearer what function in assignR was used with what inputs. For the strontium isoscape, are talking about the predicted mean and pseudo-sd

from the quantile random forest regression? Correct? If yes it needs to be specified. For the oxygen, the mean precipitation $\delta^{18}\text{O}$ values and its associated uncertainty +1 per mile? Correct? Please detail this a bit more, what function was used and with what tuning.

Response: Thank you for your suggestion. To fully address your open questions, we rewrote sections and added more details about the geographic assignments and the specific isoscapes, error maps and functions used in the “assignR” package (see lines 494-520).

“We estimated the probable geographic origins of ten enslaved Africans from two slave cemeteries in the African Diaspora. To achieve this, we applied a predicted strontium isotope of sub-Saharan Africa (mean values) and its estimated standard error (calculated using the infinitesimal jackknife) resulting from the random forest regression and the continuous-surface assignment framework from the R package “assignR” using the ‘pdRaster’ function⁷⁰. The resulting probability surfaces for each sample were normalized for comparison (each cell probability was divided by the maximum probability).

The human enamel $^{87}\text{Sr}/^{86}\text{Sr}$ data were sourced from previous studies, including five individuals from the Anson Street African Burial Ground (1760-1790 CE) in Charleston, USA¹⁰ and five individuals from the Pretos Novos cemetery (1769-1830 CE) in Rio de Janeiro, Brazil⁹. From both sites, we selected individuals with comparatively radiogenic $^{87}\text{Sr}/^{86}\text{Sr}$ ratios exceeding 0.720, as such ratios are certainly not found in coastal South Carolina²⁹ and are unlikely to be present in the coastal sugar plantation regions of Rio de Janeiro province.

For the Anson Street ancestors, we combined $^{87}\text{Sr}/^{86}\text{Sr}$ data with aDNA evidence to identify first-generation victims of the slave trade, using published genetic findings to narrow predictions of individual origin to specific African regions (West Africa vs. West-Central Africa)³⁴.

For the ancestors from the Pretos Novos cemetery, we employed dual-isotope ($^{87}\text{Sr}/^{86}\text{Sr}$ and $\delta^{18}\text{O}$) geographic assignments due to the availability of $\delta^{18}\text{O}$ data from human tooth enamel³⁵. Here we additionally used the modern precipitation oxygen isotope of Africa⁴¹ in the “assign R” framework. **We firstly adjusted human tooth carbonate $\delta^{18}\text{O}$ values to those of drinking water using established equations. Specifically, we converted enamel carbonate $\delta^{18}\text{O}$ values from V-PDB to V-SMOW using the equation from Coplen et al. (1983)⁷¹ ($\delta^{18}\text{O}_{\text{carb-VSMOW}} = 1.03091 \times \delta^{18}\text{O}_{\text{carb-VPDB}} + 30.91$), then converted carbonate $\delta^{18}\text{O}$ values to phosphate $\delta^{18}\text{O}$ values using the equation from Iacumin et al. (1996)⁷² ($\delta^{18}\text{O}_{\text{p-VSMOW}} = 0.98 \times \delta^{18}\text{O}_{\text{carb-VSMOW}} - 8.5$), and finally applied the equation from Daux et al. (2008)⁷³ to convert the resulting $\delta^{18}\text{O}$ values to drinking water $\delta^{18}\text{O}$ values ($\delta^{18}\text{O}_{\text{w-VSMOW}} = 1.54 \times \delta^{18}\text{O}_{\text{p-VSMOW}} - 33.72$). We compared the calculated human drinking water $\delta^{18}\text{O}$ values with the annual mean precipitation oxygen isotope of Africa (RCWIP)⁴¹, accounting for the uncertainty of the isotope and an additional uncertainty of 1‰ (considering the error ranges of the conversions above⁷⁴).**”

Comment 3:

2) I am not fully convinced about their reply on pseudoreplication. I totally agree with the nice new paragraph about how local uncertainty associated with the range of bioavailable $^{87}\text{Sr}/^{86}\text{Sr}$ values is meaningful and needs to be incorporated into the modeling one way or another. However, if you

take multiple points per grid cell it creates pseudo-replication by definition even if the values are slightly different. I don't have a clear solution to this problem but I did not fully understand their reply. So I am not asking the author to redo any modeling but I would like to understand why Zuckerberg, B. et al. suggest that resampling at one site does not involve pseudoreplication as this is the study cited to justify this statement? And how this cross-correlogram from Bjørnstad, O. N. & Falck, W. help us to see that there is no spatial-autocorrelation linked to pseudo-replication? Could you clarify those two points with a bit more details in the text?

Response: Thank you for your insightful comment. We are suggesting a resolution to the question of pseudoreplication v. data independence in our approach, by rigorously and transparently testing if the largest error source, spatial autocorrelation, is indeed introduced by repeated sampling from the same grid cells. We clarified this point by elaborating more on the issue of landscape resampling and the question on spatial autocorrelation in the method section describing the isoscape modeling approach (see lines 457-479):

“Unlike previous isoscape studies^{17,18}, we did not apply the median ⁸⁷Sr/⁸⁶Sr ratios for data points within the same grid cell (here 1 km²) but instead treated all data as independent. The rationale for using the median is that samples in close proximity (within the same grid cell) have the same predictor values, i.e. landscape overlap⁶⁴, and this leads to pseudoreplication and violates the assumption of independence. Firstly, we argue that our sampling is sufficiently independent, that the “resampling” is caused by low predictor resolution, and that aggregating data leads to other types of error. In fact, our ⁸⁷Sr/⁸⁶Sr samples were collected from various organisms and substrates, often hundreds of meters apart. Yet, the lack of higher-resolution maps for predictor variables at the continental scale results in these samples originating from the same cell and, thus the same predictor space (or landscape). While data from the same grid cell (1km²) will share the same environmental properties (overlapping landscapes), their ⁸⁷Sr/⁸⁶Sr ratios will differ due to natural variation of ⁸⁷Sr/⁸⁶Sr between sample locations within that cell, as well as among different sample types (e.g. various plants, soils, and animals), which are influenced by various factors affecting Sr sourcing^{17,37}. Additionally, aggregating data using medians would drastically reduce the effective sample size, leading to increased Type I error⁶⁴. Treating each data point as independent allows us to integrate the natural variation in ⁸⁷Sr/⁸⁶Sr ratios within each grid cell as an important part of the algorithm training process. Secondly, we argue that the independence assumption refers to the model residuals (the part of the variance not explained by the model) rather than the sampling^{65,66}. Zuckerberg et al. (2012)⁶⁶ demonstrated that reducing landscape oversampling does not necessarily reduce spatial autocorrelation (an indication of residual dependence in spatial models). Spatial autocorrelation in the residuals would indicate that there is an effect of sample proximity effects or landscape overlap that are not explained by the model. Therefore, to determine if our approach violates statistical independence, we assessed potential spatial auto-correlation in the model residuals using a cross-correlogram⁶⁷.”

Then, per the reviewer's request, we elaborated more on how to interpret the cross-correlogram and how it demonstrates the lack of auto-correlation in our dataset in the main text result section (see lines 221-223): **“We detected no spatial autocorrelation in the model residuals, indicating**

that there are no unaccounted proximity effects (between samples) within the variance unexplained by the model (Supplementary Fig. 2).”

We further added a sentence to the figure caption of Supplementary Figure 2 to aid the reader in interpreting the cross-correlogram:

Fig. S2. Cross-correlogram showing the cross-correlation at different distances (km) between the model residuals and $^{87}\text{Sr}/^{86}\text{Sr}$ ratios. The generally low Moran's I indicate that proximity (short distance) does not increase the correlation of residuals suggesting the absence of spatial autocorrelation in the residuals. **Further, the confidence interval (grey shading) always includes zero, indicating the absence of auto-correlation.**

Comment 4:

3) I think the author could combine the last two paragraphs (applications for wildlife/application for forensics) into one concluding paragraph describing the broader implications/applications of their isoscape. I find the last two paragraphs still distract a bit from the main message on African slave origin as they are pretty long. So I would simply have one final paragraph where the authors describe briefly all the cool applications and potential solutions that this new isoscape brings to African ecology, forensics, paleoecology... Short and exciting with a few refs.

Response: Thank you for your valuable suggestion. We have combined the last two paragraphs (applications for wildlife/application for forensics) into one concise concluding paragraph to highlight the broader implications and applications of our isoscape. Please see our revisions to the discussion (lines 368-387):

“Beyond the applications in the archaeology of the transatlantic slave trade, our

strontium isoscape, integrated with other isotopic systems (e.g., oxygen, sulfur, hydrogen, and carbon), holds transformative potential for future provenance studies across multiple disciplines. In wildlife conservation, it serves as a powerful tool for identifying the origins of traded wildlife, such as illegally logged timber⁴⁶ and endangered species⁴⁷ (e.g. elephants^{25,48,49} and chimpanzees⁵⁰), aiding efforts to combat poaching and wildlife trafficking by enabling law enforcement to pinpoint hotspots of illegal activity. Furthermore, it supports the study of wildlife dispersal (e.g. chimpanzees⁵¹) and the mobility of migratory species (e.g. bird and insect species⁵²) at large spatial scales, and provides insights into the ecology and adaptations of extinct animal species⁵³ through the analysis of field, museum, and archaeological specimens. In forensic science, our isoscape is crucial for tracing the geographic origins of unidentified human remains, particularly in the identification of deceased African migrants. For instance, in the 2001 London murder case involving the unidentified child ‘Adam’, our study suggests much broader ranges of origins within Africa than previously thought^{54,55}. Additionally, ⁸⁷Sr/⁸⁶Sr and complementary isotope systems could provide a new, rapid, and comparatively cost-effective approach to help identify the thousands of African migrants who perish in the Mediterranean Sea during their passage to southern Europe. Cattaneo and her colleagues⁵⁶ called this the “largest humanitarian disaster” in Europe since the Second World War, emphasizing the rights of the victims to be identified and repatriated if possible. Overall, these diverse applications highlight the usefulness of our isoscape, paving the way for rapid advances in provenance applications in African archaeology, ecology, forensic science, and paleoecology.”

Comment 5:

Line 173: referred to what? Weird sentence ending?

Response: Thank you for pointing this out. We revised the sentence to:

“A growing number of such studies employed strontium isotope (⁸⁷Sr/⁸⁶Sr) analysis of human remains, which demonstrates great potential for reconstructing the geographical origins and migration of individuals, particularly when local- or large-scale strontium **isoscapes are available as references**”.

Comment 6:

Remarks on code availability: I had reviewed and commented the code last time in great details. The authors have taken into account all my comments in their reviews. I did not have time to rerun fully the script this time but I believe most of the issues have been addressed.

Response: Thank you for your detailed review of our code in the previous round and for acknowledging the revisions we made. We have re-verified the code to ensure its reproducibility and reliability before publication.

Reviewer #2 (Remarks to the Author):

Comment 1:

This version of the manuscript addresses most of the problems I identified previously. My major concern about this version is to do with the interpretation of Figs 3 and 4 (see below). I also have a number of relatively minor comments, listed below.

Response: Thank you for your constructive feedback and for noting the improvements in our manuscript. We've addressed your concerns about Figures 3 and 4 in detail in the revised manuscript, and have also revised every single minor points you mentioned.

Comment 2:

First paragraph: I am not at all sure that the slave trade was “the most notable migration event” in Africa – the Bantu expansion was arguably as influential. Add “forced” before “migration”, as you say a couple of lines on, and no-one would argue.

Response: Thanks so much for your suggestion and attention to detail. We acknowledge the significance of other major migration events, such as the Bantu expansion, which we mentioned in the preceding sentence of the introduction. We have revised this sentence in the revised manuscript:

“**One of the most notable migration events in the history of the continent occurred between the 15th and 19th centuries, during which at least 12.5 million Africans were abducted, enslaved, and transported to the Americas and Europe, which dramatically changed the demography, economics, and politics of both Africa and the Americas⁵. Although the transatlantic slave trade is well documented and known as the **largest forced migration event...**”.**

Comment 3:

It is untrue that “little to nothing is known about what regions and communities they [African slaves] had been taken from” – there is a large body of scholarship on this in African history.

Response: Thank you for pointing out the vague phrasing of this sentence. We are well aware of how much we know about the transatlantic slave trade from historical documents, oral histories and even written first-hand accounts. However, we here aimed at pointing at the lack of understanding regarding the specific origins and journeys of individual persons whose skeletal remains are later found within the African Diaspora. We reworded this sentence in the introduction accordingly:

“**However, the individual treacherous journeys captive Africans endured before reaching coastal shipping ports, as well as the regions from which they were taken remained largely unclear in the bioarchaeology of the African Diaspora.**”

Comment 4:

Second paragraph: “⁸⁷Sr/⁸⁶Sr ratios in animal and human body tissues primarily mirror

biologically available $^{87}\text{Sr}/^{86}\text{Sr}$ incorporated via the consumption of plants and water from the local substrate with minimal fractionation which can be corrected during mass spectrometric analysis(Supplementary Note S1)” is still incorrect. This fraction IS corrected for (not “can be”), not during mass spectrometric analysis, rather during post-measurement data processing. These might seem like picky points, but science requires precise use of language.

Response: Thank you for your careful review and for emphasizing the importance of precise language. We appreciate your guidance on accurately describing the correction process for isotope fractionation. We have revised the sentence both in the revised manuscript (introduction) and in Supplementary Note S1:

“ $^{87}\text{Sr}/^{86}\text{Sr}$ ratios in animal and human body tissues primarily mirror biologically available $^{87}\text{Sr}/^{86}\text{Sr}$ incorporated via the consumption of plants and water from the local substrate with minimal fractionation which is corrected during post-measurement data processing^{16,17} (Supplementary Note S1).”

Comment 5:

Second paragraph of Results: Specify the “several soil properties” used as predictors in your isoscape.

Response: Thanks for your suggestion. We have now clearly listed these soil properties. Please refer to the second paragraph of the result section in the revised manuscript (lines 215-217):

“Ultimately, we used 11 predictors (dust, terrane age, **soil clay content, soil cation exchange capabilities, soil organic carbon content**, maximum age of bedrock, sea salt deposition, mean annual precipitation, srsrq3, lithology, and elevation) to model the $^{87}\text{Sr}/^{86}\text{Sr}$ ratios of 2266 samples using a random forest algorithm tuned with the hyperparameter “number of variables randomly sampled at each split” both set to 2.”

Comment 6:

It is untrue that “that RF models are unable to extrapolate into areas without matching training data”. What is true is that we can have little confidence in such extrapolations.

Response: Thank you for correcting our statement about the extrapolation capabilities of random forest models. Here is our revision to this specific sentence in the result section:

“This conservative depiction of an isoscape accounts for the fact that RF models are **not reliable when extrapolating** into areas without matching training data.”

Comment 7:

The level of “match” between $^{87}\text{Sr}/^{86}\text{Sr}$ of the skeletons and the areas they may have come from is quite uneven. Fig 3e shows very good agreement, but 3d and 3f appear to offer at best 50-60% match. Similarly, in Fig 4, 4c offers quite good agreement but 4f does not. This requires some

discussion.

Response: Thank you for raising this important point. We apologize for the lack of clarity in the original explanation of Figs. 3 and 4, which may have led to some confusion. To clarify, the probabilities shown in these figures are not the direct results from the “assignR” tool but, as we explained in the method section, normalized probability surfaces for better comparison across individuals. The non-normalized probabilities generated by assignR are presented in the Review Response Fig. 1 below (please note that the scale maximum changes between figures).

Review Response Fig. 1 The **non-normalized** probability of geographical origins for five enslaved individuals excavated from the Anson Street African Burial Ground in Charleston (USA).

In our analysis, we normalized the resulting probability surfaces for each individual by dividing the probability value of each grid cell by the maximum probability value. This means that the color scale does not represent absolute probabilities (e.g., 50-60% as you mentioned) but rather relative probabilities. As a result, darker colors indicate a higher probability compared to other cells within the same map, rather than a direct probability of origin.

For example, in Fig. 3e (Review Response Fig. 1e above), the dark color suggests that all the cells have an equal probability of being the origin, as this individual’s $^{87}\text{Sr}/^{86}\text{Sr}$ ratio (~ 0.720) is widely distributed across various regions in Africa. Therefore, many areas are equally probable

origins (with the ratio of many grid cells to the maximum probability being close to 1, see Review Response Fig. 1e). In contrast, in Figs. 3d and 3f, fewer regions show high probability because these individuals' $^{87}\text{Sr}/^{86}\text{Sr}$ ratios (> 0.730) are less common in West Africa. This leads to a more concentrated set of potential origins, where only a small number of grid cells show higher probabilities. As these grid cells are limited, the dark color may appear less prominent (see Review Response Fig. 1d and 1f).

We have modified the captions of Figs. 3 and 4 to clarify this point:

“Fig. 3 The normalized probability of geographic origins for five enslaved individuals excavated from the Anson Street African Burial Ground in Charleston (USA) using published tooth enamel $^{87}\text{Sr}/^{86}\text{Sr}$ data and our new strontium isoscape, in combination with aDNA data restricting the possible region of origin¹⁰. (a) The boxplot (representing the median and interquartile range) shows the original $^{87}\text{Sr}/^{86}\text{Sr}$ data distribution from 29 individuals buried at this site. We selected five individuals with good aDNA coverage and $^{87}\text{Sr}/^{86}\text{Sr}$ ratios ≥ 0.720 . Lower $^{87}\text{Sr}/^{86}\text{Sr}$ ratios between 0.709 and 0.715 are likely present in coastal South Carolina, where the Anson Street Ancestors were held captive and buried^{10,29}. (b) and (c) show the isotopic geolocation to parts of West-Central Africa for the individuals named *Kuto* and *Banza*, respectively. (d-f) show the isotopic geolocation of the individuals named *Daba*, *Lima*, and *Ganda* within West Africa. The probability surface was normalized by dividing each cell's value by the maximum probability. Consequently, darker regions represent areas with a higher relative probability of origin compared to other cells, rather than the actual probability values.”

“Fig. 4 The normalized probability of geographic origins for five enslaved individuals excavated from the Pretos Novos cemetery in Rio de Janeiro (Brazil) using published tooth enamel $^{87}\text{Sr}/^{86}\text{Sr}$ data⁹ and our new strontium isoscape, in combination with published enamel $\delta^{18}\text{O}$ data³⁵ and the annual mean precipitation oxygen isoscape based on RCWIP data products⁴¹. The boxplot (a) shows the original $^{87}\text{Sr}/^{86}\text{Sr}$ data distribution from 30 individuals buried at this site. From these, we selected five individuals with $^{87}\text{Sr}/^{86}\text{Sr}$ ratios ≥ 0.720 , which are very likely not present in the coastal sugar plantation regions of Rio de Janeiro province, suggesting they originated in parts of Africa with more radiogenic bedrock⁹. The isotopic geolocation suggests individual P6 (b) could have originated in West or South Africa, whereas the other four individuals (c-f) with more radiogenic $^{87}\text{Sr}/^{86}\text{Sr}$ ratios likely came from present-day Angola or the region of South-East Africa. The probability surface was normalized by dividing each cell's value by the maximum probability. Consequently, darker regions represent areas with a higher relative probability of origin compared to other cells, rather than the actual probability values.”

We hope this explanation clarifies the intent behind the figures, and we are happy to incorporate further details into the main text if needed.

Comment 8:

Does the DNA from Daba, Lima and Ganda specifically link them to the southern coast of West Africa, or just to West Africa in general? There were major West African slaving ports that fall outside the limits of this isoscape, especially Goree Island off Dakar. Could they have come from

there?

Response: Thank you for your questions. According to previous genetic studies (Fleskes et al., 2023), the aDNA from Daba, Lima, and Ganda links them to West Africa in general, not specifically to the southern coast. It is unlikely that they were from Gorée Island, as the coast of Senegal, including Gorée Island, has relatively low $^{87}\text{Sr}/^{86}\text{Sr}$ ratios (< 0.711 ; see Fig. 2a in the revised manuscript), whereas Daba, Lima, and Ganda all have $^{87}\text{Sr}/^{86}\text{Sr}$ ratios higher than 0.720. In any case, slave ports such as Gorée were only brief stations in the journey of enslaved Africans, making it highly unlikely any captive person would have $^{87}\text{Sr}/^{86}\text{Sr}$ ratios matching that of a slave port itself. People were brought to these ports from elsewhere.

From a geological perspective, Gorée Island is part of the West African Atlantic margin and features episodic volcanic activity from the Eocene-Oligocene boundary up to the Quaternary (Cap-Vert peninsula; Ndiaye et al., 2014). The island has a high zone in the south with basaltic cliffs from volcanic activity between the Oligocene and the upper Miocene, and a low zone in the north and center consisting mainly of tertiary limestone and silts (Bakhoum et al., 2017). Both basalt (0.702-0.705) and limestone (~ 0.708) typically have very low $^{87}\text{Sr}/^{86}\text{Sr}$ values (Bentley, 2006). Additionally, individuals born along the coast might also be influenced by lower marine strontium values (~ 0.7092). Therefore, we think it is unlikely that Daba, Lima, and Ganda with high $^{87}\text{Sr}/^{86}\text{Sr}$ ratios were from oceanic islands along the coast of West Africa outside of our isoscape map.

References

- Fleskes RE, Cabana GS, Gilmore JK, et al. (2023). Community-engaged ancient DNA project reveals diverse origins of 18th-century African descendants in Charleston, South Carolina. *Proceedings of the National Academy of Sciences*, 120(3): e2201620120.
- Bentley R (2006). Strontium isotopes from the earth to the archaeological skeleton: a review. *Journal of Archaeological Method and Theory*, 13(3): 135-187.
- Ndiaye A, Ngom PM (2014). The geodynamic context of the Cenozoic volcanism of the Cap-Vert Peninsula (Senegal). *International Journal of Geosciences*, 5(12): 1521.
- Bakhoum PW, Ndour A, Niang I, et al. (2017). Coastline mobility of Goree Island (Senegal), from 1942 to 2011. *Marine Science*, 7(1): 1-9.

Comment 9:

In Fig 4, what about the $\delta^{13}\text{C}$ values? Do these help narrow down the possibilities?

Response: Thank you for raising this important point. While this paper centers on strontium isotopes and a strontium isoscape, we initially considered using carbon isotopes to refine our conclusions here (and we do so in our other multi-isotope work on the archeology of the slave trade). However, several challenges limit their effectiveness in pinpointing the geographic origins of the individuals, which we re-evaluate in this study. The primary issue is the lack of a detailed crop distribution map during the time of the transatlantic slave trade. Modern C_3 and C_4 plant distributions cannot serve as accurate baselines, mainly because agricultural practices in the 16th and 19th centuries were

different.

However, some historical records indicate the regions in which the main African C₃ crops were cultivated during the time of slavery, including yam, rice, and manioc, and regions in which C₄ crops such as sorghum, millet, and maize were predominant (Ogot, 2010). In the 20th century, researchers documented the traditional crop zones and subareas in West Africa in detail (Harris, 1976), revealing that a variety of staple crops, including both C₃ and C₄ plants, were cultivated even within small regions (see details Review Response Fig. 2). This information helps narrow down the origin in the case of individual P6 (C₄ signal diet). We revised the manuscript as follows (lines 308-322):

“In the case of the Brazilian slave cemetery, Pretos Novos in Rio de Janeiro, individual P6 may have originated from West or South Africa based on the ⁸⁷Sr/⁸⁶Sr and δ¹⁸O data³⁵. This includes various discrete locations within Guinea, central Nigeria, northern Cameroon, as well as broad regions in South Africa (Fig. 4b). The C₄-plant signal in this individual’s diet might further exclude the possibility that this person originated from regions dominated by C₃ crops in West Africa, such as the “Rice Coast” - the traditional rice-growing region between Guinea and Guinea-Bissau and the western Ivory Coast - as well as the mixed vegetational zone in most regions of southern West Africa, which were dominated by yams, manioc, and other C₃ root crops (Supplementary Fig. 7)⁴³. Therefore, within West Africa, the origin of this person can be constrained to a very limited area in central Nigeria, within the sorghum-millet zone (C₄ crops), or northern Cameroon, located at the boundary between sorghum-millet and maize-dominated zones (C₄ crops). While it remains a strong possibility that individual P6 came from South Africa, which has dry farming conditions suitable for C₄ plant growing, historic records suggest a much higher probability of a West African origin, as Brazil received at least 1,540,113 captives from the West African coasts throughout the duration of the slave trade compared to 336,896 captives from South-East Africa and the Indian Ocean⁵. Future aDNA analysis could help distinguish between West and South African origins for individual P6.”

For the other four individuals with a high likelihood of origin in Angola, further narrowing down their origins is challenging. This difficulty arises from the widespread cultivation of both C₃ and C₄ crops by agricultural societies of West and Atlantic Central Africa (including Angola), following the introduction of new crop species by European colonizers (Isichei, 1997). These four individuals may also have a (less) likely origin from South-East Africa based on Sr isotopic evidence, where the tribes cultivated yam (C₃), sorghum (C₄), millet (C₄), manioc (C₃), and maize (C₄) (Ogot, 2010; Bastos et al., 2016). However, apart from sporadic historical records of agricultural activities, we currently don’t have detailed agricultural distributions within Africa from the 18th and 19th centuries. If we refer to 2005 distribution map of main crops across sub-Saharan Africa (Review Response Fig. 3), it shows that within Angola, C₃ crops (mostly manioc) dominated central and northern regions, while C₄ crops (maize and millet) were grown in central and southern regions. Maize was mainly cultivated in South-East Africa in 2005. However, the applicability of this modern map to the colonial period is uncertain, which is why we did not use carbon isotopes to further constrain the origins of these four individuals. Therefore, in the revised manuscript, when discussing these four individuals, we can only briefly mention the agricultural activities during the colonial period, as referenced in lines 328-331:

“Indeed, their tooth enamel $\delta^{13}\text{C}$ values suggest that they did not share the same dietary customs in early life. “This finding is consistent with historic records indicating that the widespread cultivation of both C_3 (manioc and other C_3 root crops) and C_4 (maize and millet) crops by tribes across Atlantic Central Africa (including Angola), introduced by European colonizers, became key staples for local populations⁴⁴.”

Additionally, apart from cultivated crops, the distribution of modern natural vegetation in Africa (Review Response Fig. 4) shows that C_4 vegetation is mainly found in the drier regions of southern Angola, while C_3 plants are widely distributed in most regions of Angola. In South-East Africa, both C_3 and C_4 vegetation are extensively distributed. This variation in natural vegetation, differing from anthropogenic crop cultivation, adds complexity to the interpretation of carbon isotope data.

Review Response Fig. 2 West Africa: traditional crop zones and subareas of crop dominance (adopted from Harris, 1976)

Review Response Fig. 3 Distribution of main crops across sub-Saharan Africa in 2005 (adopted from Fatema and Kibriya, 2019)

Review Response Fig. 4 (a) Stable carbon isotopic distribution for the African continent today, (b) percentage of vegetation that uses the C₄ pathway, and (c) percentage of vegetation that uses the C₃ pathway (adopted from Still and Powell, 2010)

References

- Isichei E (1997). *A History of African Societies to 1870*. Cambridge University Press, Cambridge.
- Fatema N, Kibriya S (2019). *Rice, Culture and conflict in sub-Saharan Africa*. Available at SSRN: <https://ssrn.com/abstract=3667213>.
- Harris DR (1976). Traditional systems of plant food production and the origins of agriculture in West Africa. In: Harlan, J, De Wet, J, Stemler, A (eds). *Origins of African plant domestication*. Mouton, pp. 311-356.
- Ogot BA (2010). *História geral da África, V: África do século XVI ao XVIII*. Unesco.
- Bastos M, Santos, R, de Souza S, et al. (2016). Isotopic study of geographic origins and diet of enslaved Africans buried in two Brazilian cemeteries. *Journal of Archaeological Science*, 70: 82-90.
- Still CJ, Powell RL (2010). Continental-scale distributions of vegetation stable carbon isotope ratios. In: West J, Bowen G, Dawson T, Tu K (eds). *Isoscapes: understanding movement, pattern, and process on Earth through isotope mapping*. Springer Netherlands, Dordrecht, pp. 179-193,

Comment 10:

Note that St Helena was simply a pick-up point – slaves taken on board there could have come from many areas. The same is true of Mozambique- slaves were brought from the interior, likely spanning several contemporary countries, and shipped from Mozambique. Individuals recorded as “of Mozambique” did therefore not necessarily originate from what we call Mozambique today.

Response: Thank you for your insightful comment. We fully agree and acknowledge that St. Helena and Mozambique were significant transit points and embarking areas respectively, with slaves taken on board potentially originating from various regions across Africa. This complexity underscores the limitations of using historical records alone to determine individual origins, which is why our strontium isotope research is crucial. As a side note, the first and last authors of this manuscript have been part of a project studying the origins of the so called “liberated” Africans from St. Helena using enamel Sr isotope data (Wang, Watson et al, in prep). Interestingly, we identified several individuals from St. Helena who likely originated from the hinterlands of the Mozambique coastline, a large region that, as shown in the current manuscript, exhibits highly radiogenic Sr isotope ratios.

We have revised the corresponding sentence in the discussion (lines 362-365) to:

“While Namibia and the Sahel, with their low population densities, were probably not the focus of slave traders⁵⁸, **the region of Mozambique and its adjacent hinterlands** were heavily impacted during the slave trade of the early to mid-19th century, with at least half a million enslaved people taken from the South-East African region⁵.”

Comment 11:

Under “Limitations and future work”, the phrase “regions with strict data extrapolation” needs rewording, making it clear that these are regions lacking actual analyses.

Response: Thanks for your suggestion. We have revised the phrase to “We employed a conservative approach by identifying and excluding **predictor space from our isoscape that would have required extrapolation.**”

Point-by-Point Response to Reviewers – Author response in blue font throughout

(please note: Line numbers refer to the numbering in the clean version of the manuscript)

Reviewers' comments:

Reviewer #1 (Remarks to the Author):

Comment 1:

Reviewer #1 (Remarks to the Author):

The authors have addressed all my comments.

- 1) I appreciate the detailed answer regarding pseudoreplication and the cross-correlogram.
- 2) I think the methods is improved with the new additions. I do think that the oxygen isoscape uncertainty could have been generated potentially in a better way by compiling oxygen data in enamel and then using `calRaster()` to generate a true uncertainty layer. However, I believe the author approach with a 1 per mile uncertainty addition to the precipitation layer is conservative.
- 3) The implications section of the manuscript is much improved.

So overall I believe that manuscript should be published.

Reviewer #1 (Remarks on code availability):

I had reviewed the code in details previously and provided detailed comments about how to make it more reproducible. I believe this has all been addressed.

Response: We appreciate the reviewer's suggestions, which greatly helped to improve the manuscript. We agree that the 1‰ uncertainty added to the precipitation layer is conservative. Given the many factors influencing oxygen isotopes (e.g., climate variability and changes in conversion formulas), it is difficult to precisely quantify the uncertainty. As a result, we adopted a conservative estimate of 1‰, based on Chenery et al. (2012), which accounts for the error ranges in the isotope conversion process.

References

Chenery CA, Pashley V, Lamb AL et al. (2012). The oxygen isotope relationship between the phosphate and structural carbonate fractions of human bioapatite. *Rapid Communications in Mass Spectrometry*, 26: 309-319.

Reviewer #2 (Remarks to the Author):

Comment 1:

I am satisfied that this revision deals with all my earlier questions and criticisms. Two very minor points:

It is inconsistent to say in line 162 that “the $^{87}\text{Sr}/^{86}\text{Sr}$ ratios of a location primarily relate to the underlying bedrock geology” and in lines 245-6 that “dust deposition is the most important factor influencing spatial bioavailable $^{87}\text{Sr}/^{86}\text{Sr}$ variation”

Response: Thank you for pointing out the inconsistency regarding the influence of strontium isotopes. As you noted, strontium isotopes are primarily known to be influenced by bedrock geology but we increasingly understand that other environmental factors might play a bigger role than we thought even just a decade ago. We revised the respective sentences for clarity.

In the Introduction section, we have changed the wording to: **“The $^{87}\text{Sr}/^{86}\text{Sr}$ ratios of a location primarily relate to the underlying bedrock geology, with additional influences from atmospheric deposition, geomorphological and biochemical process, and remain stable over archaeological time scales¹⁹.”**

In Supplementary Note 1, we have clarified further: **“While largely derived from the underlying bedrock geology, the isotopic compositions of bioavailable Sr and living organisms (e.g. plants and snail shells) can differ from the $^{87}\text{Sr}/^{86}\text{Sr}$ ratios of whole rocks and bulk soils^{7,8}. This is because some factors can also potentially influence the biosphere $^{87}\text{Sr}/^{86}\text{Sr}$ in a given region, including differential mineral compositions and weathering rates of different rock types, topographic processes, as well as non-geological sources from atmospheric deposition (e.g., rainfall, dust, and coastal sea-spray) and anthropogenic pollution⁹⁻¹¹.”**

Comment 2:

Line 422: “board” should be “broad”

Response: We have corrected the typo from “board” to “broad”.

Comment 3:

Once the authors have dealt with these, I believe the manuscript should be published.

Response: Thank you once again for your thorough and constructive comments.